# CoST: Contrastive Learning of Disentangled Seasonal-Trend Representations for Time Series Forecasting

**Gerald Woo**[1][2]**, Chenghao Liu**[1] *****, Doyen Sahoo**[1]**, Akshat Kumar**[2] **& Steven Hoi**[1]
[1]Salesforce Research Asia, [2]Singapore Management University
{gwoo,chenghao.liu,dsahoo,shoi}@salesforce.com, akshatkumar@smu.edu.sg

## ABSTRACT

Deep learning has been actively studied for time series forecasting, and the mainstream paradigm is based on the end-to-end training of neural network architectures, ranging from classical LSTM/RNNs to more recent TCNs and Transformers. Motivated by the recent success of representation learning in computer vision and natural language processing, we argue that a more promising paradigm for time series forecasting, is to first learn disentangled feature representations, followed by a simple regression fine-tuning step – we justify such a paradigm from a causal perspective. Following this principle, we propose a new time series representation learning framework for long sequence time series forecasting named CoST, which applies contrastive learning methods to learn disentangled seasonal-trend representations. CoST comprises both time domain and frequency domain contrastive losses to learn discriminative trend and seasonal representations, respectively. Extensive experiments on real-world datasets show that CoST consistently outperforms the state-of-the-art methods by a considerable margin, achieving a 21.3% improvement in MSE on multivariate benchmarks. It is also robust to various choices of backbone encoders, as well as downstream regressors. Code is available at https://github.com/salesforce/CoST.

## 1 INTRODUCTION

Time series forecasting has been widely applied to various domains, such as electricity pricing (Cuaresma et al., 2004), demand forecasting (Carbonneau et al., 2008), capacity planning and management (Kim, 2003), and anomaly detection (Laptev et al., 2017). Recently, there has been a surge of efforts applying deep learning for forecasting (Wen et al., 2017; Bai et al., 2018; Zhou et al., 2021), and owing to the increase in data availability and computational resources, these approaches have offered promising performance over conventional methods in forecasting literature. Compared to conventional approaches, these methods are able to jointly learn feature representations and the prediction function (or forecasting function) by stacking a series of non-linear layers to perform feature extraction, followed by a regression layer focused on forecasting.

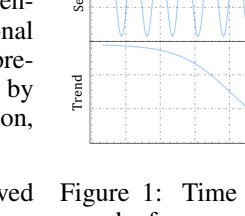

Figure 1: Time series composed of seasonal and trend components.

However, jointly learning these layers end-to-end from observed data may lead to the model over-fitting and capturing spurious correlations of the unpredictable noise contained in the observed data. The situation is exacerbated when the learned representations are entangled – when a single dimension of the feature representation encodes information from multiple local independent modules of the data-generating process – and a local independent module experiences a distribution shift. Figure 1 is an example of such a case, where the observed time series is generated by a seasonal module and nonlinear trend module. If we know that the seasonal

---

*Corresponding author.

module has experienced a distribution shift, we could still makes a reasonable prediction based on the invariant trend module. However, if we learn an entangled feature representation from the observed data, it would be challenging for the learned model to handle this distribution shift, even if it only happens in a local component of the data-generating process. In summary, the learned representations and prediction associations from the end-to-end training approach are unable to transfer nor generalize well when the data is generated from a non-stationary environment, a very common scenario in the time series analysis. Therefore, in this work, we take a step back and aim to learn disentangled seasonal-trend representations which are more useful for time series forecasting.

To achieve this goal, we leverage the idea of structural time series models (Scott & Varian, 2015; Qiu et al., 2018), which formulates time series as a sum of trend, seasonal and error variables, and exploit such prior knowledge to learn time series representations. First, we present the necessity of learning disentangled seasonal-trend representations through a causal lens, and demonstrate that such representations are robust to interventions on the error variable. Then, inspired by Mitrovic et al. (2020), we propose to simulate interventions on the error variable via data augmentations and learn the disentangled seasonal-trend representations via contrastive learning.

Based on the above motivations, we propose a novel contrastive learning framework to learn disentangled seasonal-trend representations for the Long Sequence Time-series Forecasting (LSTF) task (Zhou et al., 2021). Specifically, CoST leverages inductive biases in the model architecture to learn disentangled seasonal-trend representations. CoST efficiently learns trend representations, mitigating the problem of lookback window selection by introducing a mixture of auto-regressive experts. It also learns more powerful seasonal representations by leveraging a learnable Fourier layer which enables intra-frequency interactions. Both trend and seasonal representations are learned via contrastive loss functions. The trend representations are learned in the time domain, whereas the seasonal representations are learned via a novel frequency domain contrastive loss which encourages discriminative seasonal representations and side steps the issue of determining the period of seasonal patterns present in the data. The contributions of our work are as follows:

1. We show via a causal perspective, the benefits of learning disentangled seasonal-trend representations for time series forecasting via contrastive learning.

2. We propose CoST, a time series representation learning approach which leverages inductive biases in the model architecture to learn disentangled seasonal and trend representations, as well as incorporating a novel frequency domain contrastive loss to encourage discriminative seasonal representations.

3. CoST outperforms existing state-of-the-art approaches by a considerable margin on real-world benchmarks – 21.3% improvement in MSE for the multivariate setting. We also analyze the benefits of each proposed module, and establish that CoST is robust to various choices of backbone encoders and downstream regressors via extensive ablation studies.

## 2 SEASONAL-TREND REPRESENTATIONS FOR TIME SERIES

**Problem Formulation**   Let $(\boldsymbol{x}_1, \ldots \boldsymbol{x}_T) \in \mathbb{R}^{T \times m}$ be a time series, where $m$ is the dimension of observed signals. Given lookback window $h$, our goal is to forecast the next $k$ steps, $\hat{\boldsymbol{X}} = g(\boldsymbol{X})$, where $\boldsymbol{X} \in \mathbb{R}^{h \times m}, \hat{\boldsymbol{X}} \in \mathbb{R}^{k \times m}$, and $g(\cdot)$ denotes the prediction mapping function, and $\hat{\boldsymbol{X}}$ predicts the next $k$ time steps of $\boldsymbol{X}$.

In this work, instead of jointly learning the representation and prediction association through $g(\cdot)$, we focus on learning feature representations from observed data, with the goal of improving predictive performance. Formally, we aim to learn a nonlinear feature embedding function $\boldsymbol{V} = f(\boldsymbol{X})$, where $\boldsymbol{X} \in \mathbb{R}^{h \times m}$ and $\boldsymbol{V} \in \mathbb{R}^{h \times d}$, to project $m$-dimensional raw signals into a $d$-dimensional latent space for each timestamp. Subsequently, the learned representation of the final timestamp $\boldsymbol{v}_h$ is used as inputs for the downstream regressor of the forecasting task.

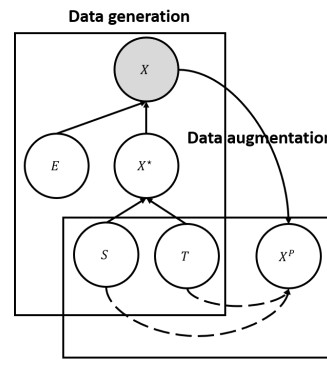

Figure 2: Causal graph of the generative process for time series data.

**Disentangled Seasonal-Trend Representation Learning and Its Causal Interpretation** As discussed in Bengio et al. (2013), complex data arise from the rich interaction of multiple sources – a good representation should be able to disentangle the various explanatory sources, making it robust to complex and richly structured variations. Not doing so may otherwise lead to capturing spurious features that do not transfer well under non i.i.d. data distribution settings.

To achieve this goal, it is necessary to introduce structural priors for time series. Here, we borrow ideas from Bayesian Structural Time Series models (Scott & Varian, 2015; Qiu et al., 2018). As illustrated in the causal graph in Figure 2, we assume that the observed time series data $X$ is generated from the error variable $E$ and the error-free latent variable $X^\star$. $X^\star$ in turn, is generated from the trend variable $T$ and seasonal variable $S$. As $E$ is not predictable, the optimal prediction can be achieved if we are able to uncover $X^\star$ which only depends on $T$ and $S$.

Firstly, we highlight that existing work using end-to-end deep forecasting methods to directly model the time-lagged relationship and the multivariate interactions along the observed data $X$. Unfortunately, each $X$ includes unpredictable noise $E$, which might lead to capturing spurious correlations. Thus, we aim to learn the error-free latent variable $X^\star$.

Secondly, by the independent mechanisms assumption (Peters et al., 2017; Parascandolo et al., 2018), we can see that the seasonal and trend modules do not influence or inform each other. Therefore, even if one mechanism changes due to a distribution shift, the other remains unchanged. The design of disentangling seasonality and trend leads to better transfer, or generalization in non-stationary environments. Furthermore, independent seasonal and trend mechanisms can be learned independently and be flexibly re-used and re-purposed.

We can see that interventions on $E$ does not influence the conditional distribution $P(X^\star|T, S)$, i.e. $P^{do(E=e_i)}(X^\star|T, S) = P^{do(E=e_j)}(X^\star|T, S)$, for any $e_i, e_j$ in the domain of $E$. Thus, $S$ and $T$ are invariant under changes in $E$. Learning representations for $S$ and $T$ allows us to find a stable association with the optimal prediction (of $X^\star$) in terms of various types of errors. Since the targets $X^\star$ are unknown, we construct a proxy contrastive learning task inspired by Mitrovic et al. (2020). Specifically, we use data augmentations as interventions on the error $E$ and learn invariant representations of $T$ and $S$ via contrastive learning. Since it is impossible to generate all possible variations of errors, we select three typical augmentations: scale, shift and jitter, which can simulate a large and diverse set of errors, beneficial for learning better representations.

## 3 SEASONAL-TREND CONTRASTIVE LEARNING FRAMEWORK

In this section, we introduce our proposed CoST framework to learn disentangled seasonal-trend representations. We aim to learn representations such that for each time step, we have the disentangled representations for seasonal and trend components, i.e., $\boldsymbol{V} = [\boldsymbol{V}^{(T)}; \boldsymbol{V}^{(S)}] \in \mathbb{R}^{h \times d}$, where $\boldsymbol{V}^{(T)} \in \mathbb{R}^{h \times d_T}$ and $\boldsymbol{V}^{(S)} \in \mathbb{R}^{h \times d_S}$, such that $d = d_T + d_S$.

Figure 3a illustrates our overall framework. Firstly, we make use of an encoder backbone $f_b : \mathbb{R}^{h \times m} \to \mathbb{R}^{h \times d}$ to map the observations to latent space. Next, we construct both the trend and seasonal representations from these intermediate representations. Specifically, the Trend Feature Disentangler (TFD), $f_T : \mathbb{R}^{h \times d} \to \mathbb{R}^{h \times d_T}$, extracts the trend representations via a mixture of auto-regressive experts and is learned via a time domain contrastive loss $\mathcal{L}_{\text{time}}$. The Seasonal Feature Disentangler (SFD), $f_S : \mathbb{R}^{h \times d} \to \mathbb{R}^{h \times d_S}$, extracts the seasonal representations via a learnable Fourier layer and is learned by a frequency domain contrastive loss which includes an amplitude component, $\mathcal{L}_{\text{amp}}$, and a phase component, $\mathcal{L}_{\text{phase}}$. We give a detailed description of both components in the next section. The model is learned in an end-to-end fashion, with the overall loss function being

$$\mathcal{L} = \mathcal{L}_{\text{time}} + \frac{\alpha}{2}(\mathcal{L}_{\text{amp}} + \mathcal{L}_{\text{phase}}),$$

where $\alpha$ is a hyper-parameter which balances the trade-off between trend and seasonal factors. Finally, we concatenate the outputs of the Trend and Seasonal Feature Disentanglers to obtain our final output representations.

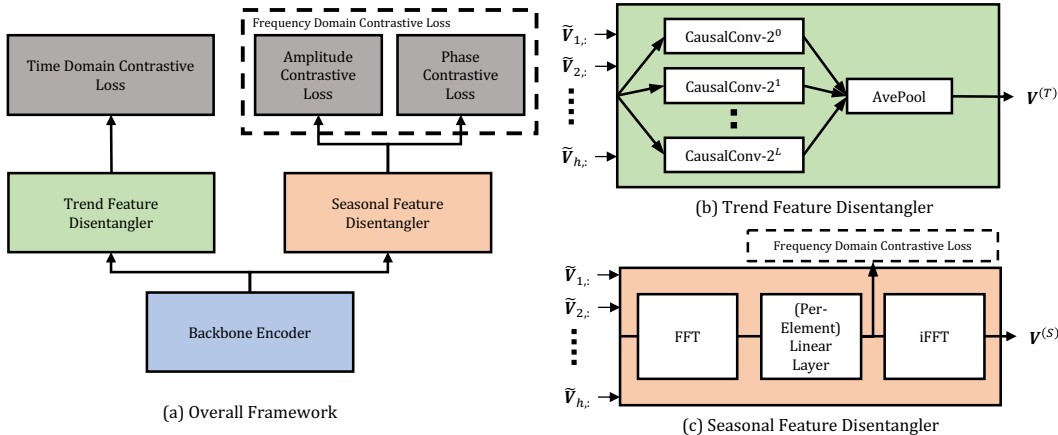

Figure 3: (a) Overall Framework. Given intermediate representations from the backbone encoder, $\tilde{\boldsymbol{V}} = f_b(\boldsymbol{X})$, the TFD and SFD produce the trend features, $\boldsymbol{V}^{(T)} = f_T(\tilde{\boldsymbol{V}})$, and seasonal features, $\boldsymbol{V}^{(S)} = f_S(\tilde{\boldsymbol{V}})$, respectively. (b) Trend Feature Disentangler. Composition of a mixture of auto-regressive experts, instantiated as 1d-causal convolutions with kernel size of $2^i, \forall i = 0, \ldots, L$, where $L$ is a hyper-parameter. Followed by average-pool over the $L+1$ representations. (c) Seasonal Feature Disentangler. After transforming the intermediate representations into frequency domain via the FFT, the SFD applies a (complex-valued) linear layer with unique weights for each frequency. Then, an inverse FFT is performed to map the representations back to time domain, to form the seasonal representations, $\boldsymbol{V}^{(S)}$.

## 3.1 TREND FEATURE REPRESENTATIONS

Extracting the underlying trend is crucial for modeling time series. Auto-regressive filtering is one widely used method, as it is able to capture time-lagged causal relationships from past observations. One challenging problem is to select the appropriate lookback window – a smaller window leads to under-fitting, while a larger model leads to over-fitting and over-parameterization issues. A straightforward solution is to optimize this hyper-parameter by grid search on the training or validation loss (Hyndman & Khandakar, 2008), but such an approach is too computationally expensive. Thus, we propose to use a mixture of auto-regressive experts which can adaptively select the appropriate lookback window.

**Trend Feature Disentangler (TFD)** As illustrated in Figure 3b, the TFD is a mixture of $L+1$ auto-regressive experts, where $L = \lfloor log_2(h/2) \rfloor$. Each expert is implemented as a $1d$ causal convolution with $d$ input channels and $d_T$ output channels, where the kernel size of the $i$-th expert is $2^i$. Each expert outputs a matrix $\tilde{\boldsymbol{V}}^{(T,i)} = \mathrm{CausalConv}(\tilde{\boldsymbol{V}}, 2^i)$. Finally, an average-pooling operation is performed over the outputs to obtain the final trend representations,

$$\boldsymbol{V}^{(T)} = \mathrm{AvePool}(\tilde{\boldsymbol{V}}^{(T,0)}, \tilde{\boldsymbol{V}}^{(T,1)}, \ldots, \tilde{\boldsymbol{V}}^{(T,L)}) = \frac{1}{(L+1)} \sum_{i=0}^{L} \tilde{\boldsymbol{V}}^{(T,i)}.$$

**Time Domain Contrastive Loss** We employ a contrastive loss in the time domain to learn discriminative trend representations. Specifically, we apply the MoCo (He et al., 2020) variant of contrastive learning which makes use of a momentum encoder to obtain representations of the positive pair, and a dynamic dictionary with a queue to obtain negative pairs. We elaborate further on the details of contrastive learning in Appendix A. Then, given $N$ samples and $K$ negative samples, the time domain contrastive loss is

$$\mathcal{L}_{\mathrm{time}} = \sum_{i=1}^{N} -\log \frac{\exp(\boldsymbol{q}_i \cdot \boldsymbol{k}_i / \tau)}{\exp(\boldsymbol{q}_i \cdot \boldsymbol{k}_i / \tau) + \sum_{j=1}^{K} \exp(\boldsymbol{q}_i \cdot \boldsymbol{k}_j / \tau)},$$

where given a sample $\boldsymbol{V}^{(T)}$, we first select a random time step $t$ for the contrastive loss and apply a projection head, which is a one-layer MLP to obtain $\boldsymbol{q}$, and $\boldsymbol{k}$ is respectively the augmented version of the corresponding sample from the momentum encoder/dynamic dictionary.

## 3.2 SEASONAL FEATURE REPRESENTATIONS

Spectral analysis in the frequency domain has been widely used in seasonality detection (Shumway et al., 2000). Thus, we turn to the frequency domain to handle the learning of seasonal representations. To do so, we aim to address two issues: i) how can we support intra-frequency interactions (between feature dimensions) which allows the representations to encode periodic information more easily, and, ii) what kind of learning signal is required to learn representations which are able to discriminate between different seasonality patterns? Standard backbone architectures are unable to easily capture intra-frequency level interactions, thus, we introduce the SFD which makes use of a learnable Fourier layer. Then, in order to learn these seasonal features without prior knowledge of the periodicity, a frequency domain contrastive loss is introduced for each frequency.

**Seasonal Feature Disentangler (SFD)** As illustrated in Figure 3c, the SFD is primarily composed of a discrete Fourier transform (DFT) to map the intermediate features to frequency domain, followed by a learnable Fourier layer. We include further details and definitions of the DFT in Appendix B. The DFT is applied along the temporal dimension and maps the time domain representations into the frequency domain, $\mathcal{F}(\tilde{\boldsymbol{V}}) \in \mathbb{C}^{F \times d}$, where $F = \lfloor h/2 \rfloor + 1$ is the number of frequencies. Next, the learnable Fourier layer, which enables frequency domain interactions, is implemented via a per-element linear layer. It applies an affine transform on each frequency, with a unique set of complex-valued parameters for each frequency, since we do not expect this layer to be translation invariant. Finally, we transform the representation back to time domain using an inverse DFT operation.

The final output matrix of this layer is the seasonal representation, $\boldsymbol{V}^{(S)} \in \mathbb{R}^{h \times d_S}$. Formally, we can denote the $i, k$-th element of the output as

$$V_{i,k}^{(S)} = \mathcal{F}^{-1}\Big(\sum_{j=1}^{d} \boldsymbol{A}_{i,j,k} \mathcal{F}(\tilde{\boldsymbol{V}})_{i,j} + B_{i,k}\Big),$$

where $\boldsymbol{A} \in \mathbb{C}^{F \times d \times d_S}, \boldsymbol{B} \in \mathbb{C}^{F \times d_S}$ are the parameters of the per-element linear layer.

**Frequency Domain Contrastive Loss** As illustrated in Figure 3c, the inputs to the frequency domain loss functions are the pre-iFFT representations, denoted by $\boldsymbol{F} \in \mathbb{C}^{F \times d_S}$. These are complex-valued representations in the frequency domain. To learn representations which are able to discriminate between different seasonal patterns, we introduce a frequency domain loss function. As our data augmentations can be interpreted as interventions on the error variable, the seasonal information does not change and thus, a contrastive loss in frequency domain corresponds to discriminating between different periodic patterns given a frequency. To overcome the issue of constructing a loss function with complex-valued representations, each frequency can be uniquely represented by its amplitude and phase representations, $|\boldsymbol{F}_{i,:}|$ and $\phi(\boldsymbol{F}_{i,:})$. Then, the loss functions are denoted,

$$\mathcal{L}_{\text{amp}} = \frac{1}{FN} \sum_{i=1}^{F} \sum_{j=1}^{N} -\log \frac{\exp(|\boldsymbol{F}_{i,:}^{(j)}| \cdot |(\boldsymbol{F}_{i,:}^{(j)})'|)}{\exp(|\boldsymbol{F}_{i,:}^{(j)}| \cdot |(\boldsymbol{F}_{i,:}^{(j)})'|) + \sum_{k \neq j}^{N} \exp(|\boldsymbol{F}_{i,:}^{(j)}| \cdot |\boldsymbol{F}_{i,:}^{(k)}|)},$$

$$\mathcal{L}_{\text{phase}} = \frac{1}{FN} \sum_{i=1}^{F} \sum_{j=1}^{N} -\log \frac{\exp(\phi(\boldsymbol{F}_{i,:}^{(j)}) \cdot \phi((\boldsymbol{F}_{i,:}^{(j)})'))}{\exp(\phi(\boldsymbol{F}_{i,:}^{(j)}) \cdot \phi((\boldsymbol{F}_{i,:}^{(j)})')) + \sum_{k \neq j}^{N} \exp(\phi(\boldsymbol{F}_{i,:}^{(j)}) \cdot \phi(\boldsymbol{F}_{i,:}^{(k)}))},$$

where $\boldsymbol{F}_{i,:}^{(j)}$ is the $j$-th sample in a mini-batch, and $(\boldsymbol{F}_{i,:}^{(j)})'$ is the augmented version of that sample.

## 4 EXPERIMENTS

In this section, we report the results of a detailed empirical analysis of CoST and compare it against a diverse set of time series representation learning approaches, as well as compare against end-to-end supervised forecasting methods. Appendix F contains further results on runtime analysis.

### 4.1 EXPERIMENTAL SETUP

**Datasets** We conduct extensive experiments on five real-world public benchmark datasets. ETT (Electricity Transformer Temperature)[1] (Zhou et al., 2021) consists of two hourly-level datasets (ETTh) and one 15-minute-level dataset (ETTm), measuring six power load features and "oil temperature", the chosen target value for univariate forecasting. Electricity[2] measures the electricity consumption of 321 clients, and following popular benchmarks, we convert the dataset into hourly-level measurements and set "MT_320" as the target value for univariate forecasting. Weather[3] is an hourly-level dataset containing 11 climate features from nearly 1,600 locations in the U.S., and we take "wet bulb" as the target value for univariate forecasting. Finally, we also include the M5 dataset (Makridakis et al., 2020) in Appendix J.

**Evaluation Setup** Following prior work, we perform experiments on two settings – multivariate and univariate forecasting. The multivariate setting involves multivariate inputs and outputs, considering all dimensions of the dataset. The univariate setting involves univariate inputs and outputs, which are the target values described above. We use MSE and MAE as evaluation metrics, and perform a 60/20/20 train/validation/test split. Inputs are zero-mean normalized and evaluated over various prediction lengths. Following (Yue et al., 2021), self-supervised learning approaches are first trained on the train split, and a ridge regression model is trained on top of the learned representations to directly forecast the entire prediction length. The validation set is used to choose the appropriate ridge regression regularization term $\alpha$, over a search space of $\{0.1, 0.2, 0.5, 1, 2, 5, 10, 20, 50, 100, 200, 500, 1000\}$. Evaluation results are reported on the test set.

**Implementation Details** For CoST and all other representation learning methods, the backbone encoder used is a Temporal Convolution Network (following similar practice in TS2Vec (Yue et al., 2021)) unless the approach includes or is an architectural modification (further details in Appendix E). All methods used have a representation dimensionality of 320. We use a standard hyper-parameter setting on all datasets – a batch size of 256 and learning rate of $1E{-}3$, momentum of 0.9 and weight decay of $1E{-}4$ with SGD optimizer and cosine annealing. The MoCo implementation for time domain contrastive loss uses a queue size of 256, momentum of 0.999, and temperature of 0.07. We train for 200 iterations for datasets with less than 100,000 samples, and 600 iterations otherwise. Details on data augmentations used in CoST can be found in Appendix C.

### 4.2 RESULTS

Among the baselines, we report the performance of representation learning techniques including TS2Vec, TNC, and a time series adaptation of MoCo in our main results. A more extensive benchmark of feature-based forecasting approaches can be found in Appendix H due to space limitations. Further details about the baselines can be found in Appendix E. We include supervised forecasting approaches - two Transformer based models, Informer (Zhou et al., 2021) and LogTrans(Li et al., 2020), and the backbone TCN trained directly on an end-to-end forecasting loss. A comparison of end-to-end forecasting approaches can be found in Appendix I.

Table 1 summarizes the results of CoST and top performing baselines for the multivariate setting, and Table 7 (in Appendix G due to space limitations) for the univariate setting. For end-to-end forecasting approaches, the TCN generally outperforms the Transformer based approaches, Informer and LogTrans. At the same time, the representation learning methods outperform end-to-end forecasting approaches, but there are indeed cases, such as in certain datasets for the univariate setting, where the end-to-end TCN performs surprisingly well. While Transformers have been shown to

---

[1]https://github.com/zhouhaoyi/ETDataset

[2]https://archive.ics.uci.edu/ml/datasets/ElectricityLoadDiagrams20112014

[3]https://www.ncei.noaa.gov/data/local-climatological-data/

Table 1: Multivariate forecasting results. Best results are highlighted in bold.

| Methods | | Representation Learning | | | | | | | | End-to-end Forecasting | | | | | |
|---|---|---|---|---|---|---|---|---|---|---|---|---|---|---|---|
| | | CoST | | TS2Vec | | TNC | | MoCo | | Informer | | LogTrans | | TCN | |
| Metrics | | MSE | MAE | MSE | MAE | MSE | MAE | MSE | MAE | MSE | MAE | MSE | MAE | MSE | MAE |
| ETTh1 | 24 | **0.386** | **0.429** | 0.590 | 0.531 | 0.708 | 0.592 | 0.623 | 0.555 | 0.577 | 0.549 | 0.686 | 0.604 | 0.583 | 0.547 |
| | 48 | **0.437** | **0.464** | 0.624 | 0.555 | 0.749 | 0.619 | 0.669 | 0.586 | 0.685 | 0.625 | 0.766 | 0.757 | 0.670 | 0.606 |
| | 168 | **0.643** | **0.582** | 0.762 | 0.639 | 0.884 | 0.699 | 0.820 | 0.674 | 0.931 | 0.752 | 1.002 | 0.846 | 0.811 | 0.680 |
| | 336 | **0.812** | **0.679** | 0.931 | 0.728 | 1.020 | 0.768 | 0.981 | 0.755 | 1.128 | 0.873 | 1.362 | 0.952 | 1.132 | 0.815 |
| | 720 | **0.970** | **0.771** | 1.063 | 0.799 | 1.157 | 0.830 | 1.138 | 0.831 | 1.215 | 0.896 | 1.397 | 1.291 | 1.165 | 0.813 |
| ETTh2 | 24 | 0.447 | 0.502 | **0.423** | **0.489** | 0.612 | 0.595 | 0.444 | 0.495 | 0.720 | 0.665 | 0.828 | 0.750 | 0.935 | 0.754 |
| | 48 | 0.699 | 0.637 | **0.619** | **0.605** | 0.840 | 0.716 | 0.613 | 0.595 | 1.457 | 1.001 | 1.806 | 1.034 | 1.300 | 0.911 |
| | 168 | **1.549** | **0.982** | 1.845 | 1.074 | 2.359 | 1.213 | 1.791 | 1.034 | 3.489 | 1.515 | 4.070 | 1.681 | 4.017 | 1.579 |
| | 336 | **1.749** | **1.042** | 2.194 | 1.197 | 2.782 | 1.349 | 2.241 | 1.186 | 2.723 | 1.340 | 3.875 | 1.763 | 3.460 | 1.456 |
| | 720 | **1.971** | **1.092** | 2.636 | 1.370 | 2.753 | 1.394 | 2.425 | 1.292 | 3.467 | 1.473 | 3.913 | 1.552 | 3.106 | 1.381 |
| ETTm1 | 24 | **0.246** | **0.329** | 0.453 | 0.444 | 0.522 | 0.472 | 0.458 | 0.444 | 0.323 | 0.369 | 0.419 | 0.412 | 0.363 | 0.397 |
| | 48 | **0.331** | **0.386** | 0.592 | 0.521 | 0.695 | 0.567 | 0.594 | 0.528 | 0.494 | 0.503 | 0.507 | 0.583 | 0.542 | 0.508 |
| | 96 | **0.378** | **0.419** | 0.635 | 0.554 | 0.731 | 0.595 | 0.621 | 0.553 | 0.678 | 0.614 | 0.768 | 0.792 | 0.666 | 0.578 |
| | 288 | **0.472** | **0.486** | 0.693 | 0.597 | 0.818 | 0.649 | 0.700 | 0.606 | 1.056 | 0.786 | 1.462 | 1.320 | 0.991 | 0.735 |
| | 672 | **0.620** | **0.574** | 0.782 | 0.653 | 0.932 | 0.712 | 0.821 | 0.674 | 1.192 | 0.926 | 1.669 | 1.461 | 1.032 | 0.756 |
| Electricity | 24 | **0.136** | **0.242** | 0.287 | 0.375 | 0.354 | 0.423 | 0.288 | 0.374 | 0.312 | 0.387 | 0.297 | 0.374 | 0.235 | 0.346 |
| | 48 | **0.153** | **0.258** | 0.309 | 0.391 | 0.376 | 0.438 | 0.310 | 0.390 | 0.392 | 0.431 | 0.316 | 0.389 | 0.253 | 0.359 |
| | 168 | **0.175** | **0.275** | 0.335 | 0.410 | 0.402 | 0.456 | 0.337 | 0.410 | 0.515 | 0.509 | 0.426 | 0.466 | 0.278 | 0.372 |
| | 336 | **0.196** | **0.296** | 0.351 | 0.422 | 0.417 | 0.466 | 0.353 | 0.422 | 0.759 | 0.625 | 0.365 | 0.417 | 0.287 | 0.382 |
| | 720 | **0.232** | **0.327** | 0.378 | 0.440 | 0.442 | 0.483 | 0.380 | 0.441 | 0.969 | 0.788 | 0.344 | 0.403 | 0.287 | 0.381 |
| Weather | 24 | **0.298** | **0.360** | 0.307 | 0.363 | 0.320 | 0.373 | 0.311 | 0.365 | 0.335 | 0.381 | 0.435 | 0.477 | 0.321 | 0.367 |
| | 48 | **0.359** | **0.411** | 0.374 | 0.418 | 0.380 | 0.421 | 0.372 | 0.416 | 0.395 | 0.459 | 0.426 | 0.495 | 0.386 | 0.423 |
| | 168 | **0.464** | **0.491** | 0.491 | 0.506 | 0.479 | 0.495 | 0.482 | 0.499 | 0.608 | 0.567 | 0.727 | 0.671 | 0.491 | 0.501 |
| | 336 | **0.497** | 0.517 | 0.525 | 0.530 | 0.505 | 0.514 | 0.516 | 0.523 | 0.702 | 0.620 | 0.754 | 0.670 | 0.502 | **0.507** |
| | 720 | 0.533 | 0.542 | 0.556 | 0.552 | 0.519 | 0.525 | 0.540 | 0.540 | 0.831 | 0.731 | 0.885 | 0.773 | **0.498** | **0.508** |
| Avg. | | **0.590** | **0.524** | 0.750 | 0.607 | 0.870 | 0.655 | 0.753 | 0.608 | 1.038 | 0.735 | 1.180 | 0.837 | 0.972 | 0.666 |

be powerful models in other domains like NLP, this suggests that TCN models are still a powerful baseline which should still be considered for time series.

Overall, our approach achieves state-of-the-art performance, beating the best performing end-to-end forecasting approach by 39.3% and 18.22% (MSE) in the multivariate and univariate settings respectively. CoST also beats next best performing feature-based approach by 21.3% and 4.71% (MSE) in the multivariate and univariate settings respectively. This indicates that CoST learns more relevant features by learning a composition of trend and seasonal features which are crucial for forecasting tasks.

## 4.3 PARAMETER SENSITIVITY

Table 2: Parameter sensitivity of $\alpha$ in CoST on the ETT datasets.

| $\alpha$ | | 1E-01 | 5E-02 | 1E-02 | 5E-03 | 1E-03 | 5E-04 | 1E-04 | 5E-05 | 1E-05 |
|---|---|---|---|---|---|---|---|---|---|---|
| Multivariate | | 0.810 | 0.805 | 0.781 | 0.781 | 0.782 | 0.781 | 0.780 | 0.780 | 0.780 |
| Univariate | | 0.120 | 0.113 | 0.106 | 0.104 | 0.102 | 0.102 | 0.103 | 0.103 | 0.103 |
| Cases for which larger $\alpha$ is preferred (multivariate). | | | | | | | | | | |
| ETTh2 | 168 | 1.509 | 1.604 | 1.555 | 1.550 | 1.550 | 1.549 | 1.544 | 1.545 | 1.546 |
| | 336 | 1.524 | 1.646 | 1.722 | 1.731 | 1.744 | 1.749 | 1.759 | 1.762 | 1.765 |

$\alpha$ controls the weightage of the seasonal components in the overall loss function, $\mathcal{L} = \mathcal{L}_{\text{time}} + \frac{\alpha}{2}(\mathcal{L}_{\text{amp}} + \mathcal{L}_{\text{phase}})$. We perform a sensitivity analysis on this hyper-parameter (Table 2) and show that an optimal value can be chosen and is robust across various settings. We choose $\alpha = 5\text{E}-04$ for all other experiments since it performs well on both multivariate and univariate settings. We note that the small values of $\alpha$ stems from the frequency domain contrastive loss being generally three orders of magnitude larger than the time domain contrastive loss, rather than being an indicator that the seasonal component has lower importance than the trend component. Further, we highlight that overall, while choosing a smaller value of $\alpha$ leads to better performance on most datasets, there are certain cases for which a larger $\alpha$ might be preferred, as seen in the lower portion of Table 2.

## 4.4 ABLATION STUDY

Table 3: Ablation study of various components of CoST on ETT datasets). TFD: Trend Feature Disentangler, MARE: Mixture of Auto-regressive Experts (TFD without MARE refers to the TFD module with a single AR expert with kernel size $\lfloor h/2 \rfloor$), SFD: Seasonal Feature Disentangler, LFL: Learnable Fourier Layer, FDCL: Frequency Domain Contrastive Loss. † indicates a model trained end-to-end with supervised forecasting loss. ‡ indicates † with an additional contrastive loss.

| | TFD | MARE | SFD | LFL | FDCL | Multivariate | | Univariate | |
|---|---|---|---|---|---|---|---|---|---|
| | | | | | | MSE | MAE | MSE | MAE |
| Trend | ✓ | | | | | 0.882 | 0.674 | 0.115 | 0.243 |
| | ✓ | ✓ | | | | 0.789 | 0.630 | 0.105 | 0.235 |
| Seasonal | | | ✓ | | ✓ | 0.905 | 0.675 | 0.105 | 0.237 |
| | | | ✓ | ✓ | | 0.895 | 0.721 | 0.103 | 0.239 |
| | | | ✓ | ✓ | ✓ | 0.862 | 0.668 | 0.129 | 0.255 |
| CoST† | | | - | | | 1.376 | 0.834 | 0.228 | 0.366 |
| CoST‡ | | | - | | | 1.477 | 0.909 | 0.965 | 0.883 |
| MoCo | | | - | | | 0.996 | 0.721 | 0.112 | 0.248 |
| SimCLR | | | - | | | 1.021 | 0.730 | 0.113 | 0.248 |
| CoST | ✓ | ✓ | ✓ | ✓ | ✓ | **0.781** | **0.625** | **0.102** | **0.233** |

Table 4: Ablation study of various backbone encoders on the ETT datasets.

| Backbones | TCN | | | | LSTM | | | | Transformer | | | |
|---|---|---|---|---|---|---|---|---|---|---|---|---|
| Methods | TS2Vec | | CoST | | TS2Vec | | CoST | | TS2Vec | | CoST | |
| | MSE | MAE | MSE | MAE | MSE | MAE | MSE | MAE | MSE | MAE | MSE | MAE |
| Multivariate | 0.990 | 0.717 | **0.781** | **0.625** | 1.415 | 0.903 | **0.928** | **0.706** | 1.092 | 0.766 | **0.863** | **0.674** |
| Univariate | 0.116 | 0.253 | **0.102** | **0.233** | 0.544 | 0.596 | **0.148** | **0.301** | 0.172 | 0.328 | **0.159** | **0.320** |

Table 5: Ablation study of various regressors on the ETT datasets

| Regressors | Ridge | | | | Linear | | | | Kernel Ridge | | | |
|---|---|---|---|---|---|---|---|---|---|---|---|---|
| Methods | TS2Vec | | CoST | | TS2Vec | | CoST | | TS2Vec | | CoST | |
| | MSE | MAE | MSE | MAE | MSE | MAE | MSE | MAE | MSE | MAE | MSE | MAE |
| Multivariate | 0.990 | 0.717 | **0.781** | **0.625** | 1.821 | 0.944 | **1.472** | **0.781** | 1.045 | 0.738 | **0.868** | **0.686** |
| Univariate | 0.116 | 0.253 | **0.102** | **0.233** | 0.304 | 0.414 | **0.182** | **0.310** | 0.132 | 0.273 | **0.109** | **0.243** |

**Components of CoST** We first perform an ablation study to understand the performance benefits brought by each component in CoST. Table 3 presents the average results over the ETT datasets on all forecast horizon settings (similarly for Tables 4 and 5). We show that both trend and seasonal components improve performance over the baselines (SimCLR and MoCo), and further, the composition of trend and seasonal components leads to the optimal performance. We further verify that training our proposed model architecture end-to-end with a supervised forecasting loss leads to worse performance.

**Backbones** Next, we verify that our proposed trend and seasonal components as well as contrastive loss (both time and frequency domain) are robust to various backbone encoders. TCN is the default backbone encoder used in all other experiments and we present results on LSTM and Transformer backbone encoders of equivalent parameter size. While performance using the TCN backbone outperforms LSTM and Transformer, we show that our approach outperforms the competing approach on all three settings.

**Regressors** Finally, we show that CoST is also robust to various regressors used for forecasting. Apart from a ridge regression model, we also perform experiments on a linear regression model and a kernel ridge regression model with RBF kernel. As shown in Table 5, we also demonstrate that CoST outperforms the competing baseline on all three settings.

### 4.5 CASE STUDY

We visualize the learned representations on a simple synthetic time series with both seasonal and trend components, and show that CoST is able to learn representations which are able to discriminate between various seasonal and trend patterns. The synthetic dataset is generated by defining two trend and three seasonal patterns, and taking the cross product to form six time series (details in Appendix D). After training the encoders on the synthetic dataset, we can visualize them via the T-SNE algorithm (Van der Maaten & Hinton, 2008). Figure 4 shows that our approach is able to learn both the trend and seasonal patterns from the dataset and the learned representations has high clusterability, whereas TS2Vec is unable to distinguish between various seasonal patterns.

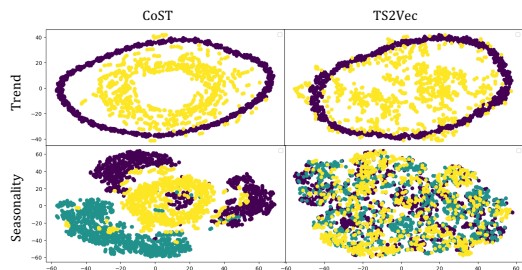

Figure 4: T-SNE visualization of learned representations from CoST and TS2Vec. (Top) Generated by visualizing representations after selecting a single seasonality. Colors represent the two distinct trends. (Bottom) Generated by visualizing representations after selecting a single trend. Colors represent the three distinct seasonal patterns.

## 5 RELATED WORK

Deep forecasting has typically been tackled as an end-to-end supervised learning task, where early work considered using RNN based models (Lai et al., 2018) as a natural approach to modeling time series data. Recent work have also considered adapting Transformer based models for time series forecasting (Li et al., 2020; Zhou et al., 2021), specifically focusing on tackling the quadratic space complexity of Transformer models. Oreshkin et al. (2020) proposed a univariate deep forecasting model and showed that deep models outperform classical time series forecasting techniques.

While recent work in time series representation learning focused on various aspects of representation learning such how to sample contrastive pairs (Franceschi et al., 2020; Tonekaboni et al., 2021), taking a Transformer based approach (Zerveas et al., 2021), exploring complex contrastive learning tasks (Eldele et al., 2021), as well as constructing temporally hierarchical representations (Yue et al., 2021), none have touched upon learning representations composed of trend and seasonal features. Whereas existing work have focused exclusively on time series classification tasks, Yue et al. (2021) first showed that time series representations learned via contrastive learning establishes a new state-of-the-art performance on deep forecasting benchmarks.

Classical time series decomposition techniques (Hyndman & Athanasopoulos, 2018) have been used to decompose time series into seasonal and trend components to attain interpretability. There has been recent work on developing more robust and efficient decomposition approaches (Wen et al., 2018; 2020; Yang et al., 2021). These methods focus decomposing the raw time series into trend and seasonal components which are still interpreted as time series in the original input space rather than learning representations. Godfrey & Gashler (2017) presents an initial attempt to use neural networks to model periodic and non-periodic components in time series data, leveraging periodic activation functions to model the periodic components. Different from our work, such a method is only able to model a single time series per model, rather than produce the decomposed seasonal-trend representations given a lookback window.

## 6 CONCLUSION

Our work shows separating the representation learning and downstream forecasting task to be a more promising paradigm than the standard end-to-end supervised training approach for time-series forecasting. We show this empirically, and also explain it through a causal perspective. By following this principle, we proposed CoST, a contrastive learning framework that learns disentangled seasonal-trend representations for time series forecasting tasks. Extensive empirical analysis shows that CoST outperforms the previous state-of-the-art approaches by a considerable margin and is robust to various choices of backbone encoders and regressors. Future work will extend our framework for other time-series intelligence tasks.

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

## A  CONTRASTIVE LEARNING

Contrastive learning via the instance discrimination task is a powerful approach for self-supervised learning. Here, we describe the essentials of this method, which underpins our proposed approach. Firstly, a family of data augmentations, $\mathcal{A}$, is defined. Given a single sample of data $\boldsymbol{x}_i \in \mathbb{X}$, two data augmentation operators are sampled, $a \sim \mathcal{A}, a' \sim \mathcal{A}$. $\boldsymbol{q}_i = f(a(\boldsymbol{x}_i))$ is referred to as the query representation with encoder $f$, and $\boldsymbol{k}_i = f(a'(\boldsymbol{x}_i))$ is the positive key representation. Finally, the InfoNCE loss function is

$$\mathcal{L}_{\text{InfoNCE}} = \sum_{i=1}^{N} - \log \frac{\exp(\boldsymbol{q}_i \cdot \boldsymbol{k}_i/\tau)}{\exp(\boldsymbol{q}_i \cdot \boldsymbol{k}_i/\tau) + \sum_{j=1}^{K} \exp(\boldsymbol{q}_i \cdot \boldsymbol{k}_j/\tau)},$$

where $\tau$ is the temperature hyper-parameter, $k_j$ are negative key representations, and $K$ is the total number of negative samples. Standard approaches use an efficient mechanism to obtain negative samples - by simply treating all other samples in the mini-batch as negative samples, i.e. $K = N-1$. MoCo (He et al., 2020) introduces the idea of a using a queue of size $K$ (a hyper-parameter) to obtain negative samples. At each iteration of training, simply pop $N$ samples from the queue, and push the $N$ representations form the current mini-batch.

## B  DISCRETE FOURIER TRANSFORM

The DFT provides a frequency domain view of a given sequence, $\boldsymbol{x} = (x_0, x_1, \ldots, x_{N-1})$, mapping a time series with regular intervals into the Fourier coefficients, a sequence of complex numbers of equal length. Due to the conjugate symmetry of the DFT of real-valued signals, we can simply consider the first $\lfloor N/2 \rfloor + 1$ Fourier coefficients, $\boldsymbol{c} = \mathcal{F}(\boldsymbol{x}) \in \mathbb{C}^{\lfloor N/2 \rfloor + 1}$.

$$c_k = \mathcal{F}(\boldsymbol{x})_k = \sum_{n=0}^{N-1} x_n \cdot \exp(-i2\pi kn/N).$$

Each complex component, $c_k$, can be represented by the amplitude, $|c_k|$, and the phase, $\phi(c_k)$,

$$|c_k| = \sqrt{\mathfrak{R}\{c_k\}^2 + \mathfrak{I}\{c_k\}^2} \qquad\qquad \phi(c_k) = \tan^{-1}\left(\frac{\mathfrak{I}\{c_k\}}{\mathfrak{R}\{c_k\}}\right)$$

where $\mathfrak{R}\{c_k\}$ and $\mathfrak{I}\{c_k\}$ are the real and imaginary components of $c_k$ respectively. Finally, the inverse DFT maps the frequency domain representation back to the time domain,

$$x_n = \mathcal{F}^{-1}(\boldsymbol{c})_n = \frac{1}{N} \sum_{k=0}^{N-1} c_k \cdot \exp(i2\pi kn/N).$$

## C  DATA AUGMENTATIONS

In our experiments, we utilize a composition of three data augmentations, applied in the following order - scaling, shifting, and jittering, activating with a probability of 0.5.

**Scaling**  The time-series is scaled by a single random scalar value, obtained by sampling $\epsilon \sim \mathcal{N}(0, 0.5)$, and each time step is $\tilde{x}_t = \epsilon x_t$.

**Shifting**  The time-series is shifted by a single random scalar value, obtained by sampling $\epsilon \sim \mathcal{N}(0, 0.5)$ and each time step is $\tilde{x}_t = x_t + \epsilon$.

**Jittering**  I.I.D. Gaussian noise is added to each time step, from a distribution $\epsilon_t \sim \mathcal{N}(0, 0.5)$, where each time step is now $\tilde{x}_t = x_t + \epsilon_t$.

## D  SYNTHETIC DATA GENERATION

We first construct two trend patterns. The first trend patterns follows a nonlinear, saturating pattern, $y_t = \frac{1}{1 + \exp \beta_0(t - \beta_1)} + \epsilon_t$ for $\beta_0 = 0.2, \beta_1 = 60, \epsilon_t \sim \mathcal{N}(0, 0.3)$. The second pattern is a mixture of

ARMA process, $\mathrm{ARMA}(2, 2) + \mathrm{ARMA}(3, 3) + \mathrm{ARMA}(4, 4)$, where the AR and MA parameters are as follows, ({.9, -.1}, {.2, -.5}), ({.1, .2, .3}, {.1, .65, -.45}), ({.3, .5, -.5, -.3}, {.1, .1, -.2, -.3}). Next, we construct three seasonal patterns, consisting of sine waves with the following period, phase, and amplitudes, {20, 0, 3}, {50, .2, 3}, {100, .5, 3}.

The final time series are constructed as follows, generate a trend pattern $g(t)$ and seasonal pattern $s(t)$, the final time series is $y(t) = g(t) + s(t), t = 0, \ldots, 999$. We do this for all pairs of trend and seasonal patterns, constructing a total of 6 time series.

## E  DETAILS ON BASELINES

Results for TS2Vec, TNC, MoCo, Triplet, CPC, TST, TCC are based on our reproduction, while results for Informer, LogTrans, and LSTNet (Lai et al., 2018) are directly taken from Yue et al. (2021) for the ETT and Electricity datasets, and Zhou et al. (2021) for the Weather dataset. For all reproduced approaches, we train for 200 iterations for datasets with less than 100,000 samples, otherwise, and 600 iterations otherwise.

The encoders used in all approaches except Triplet and TST is the causal TCN encoder as proposed in TS2Vec (Yue et al., 2021). A fully connected layer is first used to project each time step from the dimensionality of the multivariate time series to the hidden channel (size 64). Then, there are 10 layers of convolution blocks. Each convolution block is a sequence of GELU, DilatedConv, GELU, DilatedConv, with skip connections across each block. The DilatedConvs have dilation of $2^i$ in each layer $i$ of convolution block, all have kernel size of 3 and input/output channel size of 64. A final convolution block is used to map the hidden channels to the output channel (size 320).

For Triplet, the encoder is their proposed causal TCN encoder, while for TST, their proposed Transformer encoder is used. Further details can be found in their respective papers.

**TS2Vec (Yue et al., 2021)**   TS2Vec was recently proposed as a universal framework for learning time series representations by performing contrastive learning in a hierarchical manner over augmented context views. They propose to learn timestamp level representations. We ran the code from their open source repository[4] as is, hyper-parameters used are all defaults as suggested in their paper.

**TNC (Tonekaboni et al., 2021)**   TNC proposes a self-supervised framework for learning generalizable representations for non-stationary time series. They make use of the augmented Dickey-Fuller test for stationarity to ensure positive samples come from the a neighborhood of similar signals. We use their open source code[5], setting $w = 0.005$ in the loss function, $\mathrm{mc\_sample\_size} = 20$, batch size of 8, and learning rate of $1\mathrm{E}{-}3$ with Adam optimizer.

**MoCo (He et al., 2020)**   MoCo is a popular self-supervised learning baseline in computer vision, and we implement a time series version of MoCo using their open source code[6]. We use a batch size of 128, queue of 256, momentum for the momentum encoder is 0.999, temperature of the loss function is 0.07, learning rate of $1\mathrm{E}{-}3$ with SGD optimizer, with cosine annealing. Data augmentations used are described in Appendix C.

**Triplet (Franceschi et al., 2020)**   Triplet proposes a time series self-supervised learning approach by taking positive samples to be substrings of the anchor, and negative samples to be randomly sampled from the dataset. We reproduce their approach by adapting their open source code[7], making use of their proposed causal TCN model architecture. We use a batch size of 10, and learning rate $1\mathrm{E}{-}3$ with AdamW optimizer.

**CPC (van den Oord et al., 2019)**   CPC learns representations by predicting the future in latent space using auto-regressive models. They use a probabilistic contrastive loss which induces latent

---

[4]https://github.com/yuezhihan/ts2vec
[5]https://github.com/sanatonek/TNC_representation_learning
[6]https://github.com/facebookresearch/moco
[7]https://github.com/White-Link/UnsupervisedScalableRepresentationLearningTimeSeries

space to capture information maximally useful to predict future samples. We reproduce this approach by referencing an unofficial implementation[8]. We use a GRU for the AR module, use two prediction steps after eight AR steps. We use a batch size of 8 and learning rate of $1E-3$ with AdamW optimizer.

**TST (Zerveas et al., 2021)**   TST is a Transformer based approach using a reconstruction loss. We use the open source implementation[9] as is.

**TCC (Eldele et al., 2021)**   TCC introduces a tempral contrastive module and a tough cross-view prediction task. We use their open source implementation[10]. We use a learning rate of $3E-4$ with Adam optimizer $\lambda_1 = 1, \lambda_2 = 0.7$ as in their implementation, and jitter scale ratio or 1.1, maximum segment length of 8, and jitter ratio of 0.8, following their HAR setting.

## F   RUNTIME ANALYSIS

Table 6: Runtime (seconds) for each method in train and inference phase. For representation methods, we split the runtime into A + B, where A refers to the time for encoder, and B is the time for the ridge regressor in downstream phase.

| Phase | H | CoST | TS2Vec | TNC | MoCo | Informer | TCN |
|---|---|---|---|---|---|---|---|
| Training | 24 | 262.78 + 7.19 | 91.9 + 5.45 | 1801.58 + 4.78 | 31.3 + 6.57 | 331.32 | 108.36 |
| | 48 | 262.78 + 8.41 | 91.9 + 6.44 | 1801.58 + 5.79 | 31.3 + 8.03 | 167.18 | 109.30 |
| | 96 | 262.78 + 10.0 | 91.9 + 8.13 | 1801.58 + 7.23 | 31.3 + 9.40 | 325.51 | 110.02 |
| | 288 | 262.78 + 19.9 | 91.9 + 16.47 | 1801.58 + 14.51 | 31.3 + 17.83 | 449.86 | 112.08 |
| | 672 | 262.78 + 38.3 | 91.9 + 32.22 | 1801.58 + 28.21 | 31.3 + 36.63 | 587.25 | 112.87 |
| Inference | 24 | 23.60 + 0.04 | 3.73 + 0.05 | 3.38 + 0.05 | 3.67 + 0.07 | 10.32 | 3.20 |
| | 48 | 23.60 + 0.07 | 3.73 + 0.06 | 3.38 + 0.09 | 3.67 + 0.08 | 5.78 | 3.67 |
| | 96 | 23.60 + 0.11 | 3.73 + 0.08 | 3.38 + 0.09 | 3.67 + 0.10 | 11.32 | 4.78 |
| | 288 | 23.60 + 0.24 | 3.73 + 0.18 | 3.38 + 0.21 | 3.67 + 0.22 | 17.19 | 7.19 |
| | 672 | 23.60 + 0.34 | 3.73 + 0.30 | 3.38 + 0.33 | 3.67 + 0.33 | 25.62 | 11.93 |

Table 6 shows the runtime in seconds for each phase for various representation learning methods and end-to-end approaches. All experiments are performed on an NVIDIA A100 GPU. Do note that for Informer, we follow the hyperparameters (including lookback window length) as described by the authors which may vary for different forecasting horizons, thus the runtime may not be strictly increasing as the forecasting horizon increases. We want to highlight that for all representation learning approaches, the ridge regressor portions should be equal since the dimension size used are the same across all methods, any differences are simply due to randomness. Despite a slightly higher training time compared to some of the baseline approaches, CoST achieves much better results (refer to Table 1 in the main paper). Furthermore, the extra computation time of CoST as compared to TS2Vec is due to the sequential computation of each expert in the mixture of AR expert component, it can be further accelerated by parallel methods (He et al., 2021).

---

[8]https://github.com/jefflai108/Contrastive-Predictive-Coding-PyTorch

[9]https://github.com/gzerveas/mvts_transformer

[10]https://github.com/emadeldeen24/TS-TCC

# G  UNIVARIATE FORECASTING BENCHMARK

Table 7: Univariate forecasting results. Best results are highlighted in bold.

| Methods | | \multicolumn{8}{c}{Representation Learning} | | | | | | | | \multicolumn{6}{c}{End-to-end Forecasting} | | | | | | Feature Engineered | |
|---|---|---|---|---|---|---|---|---|---|---|---|---|---|---|---|---|---|---|---|
| | | CoST | | TS2Vec | | TNC | | MoCo | | Informer | | LogTrans | | TCN | | TSFresh | |
| Metrics | | MSE | MAE | MSE | MAE | MSE | MAE | MSE | MAE | MSE | MAE | MSE | MAE | MSE | MAE | MSE | MAE |
| ETTh1 | 24 | 0.040 | 0.152 | **0.039** | **0.151** | 0.057 | 0.184 | 0.040 | **0.151** | 0.098 | 0.247 | 0.103 | 0.259 | 0.104 | 0.254 | 0.080 | 0.224 |
| | 48 | **0.060** | **0.186** | 0.062 | 0.189 | 0.094 | 0.239 | 0.063 | 0.191 | 0.158 | 0.319 | 0.167 | 0.328 | 0.206 | 0.366 | 0.092 | 0.242 |
| | 168 | **0.097** | **0.236** | 0.142 | 0.291 | 0.171 | 0.329 | 0.122 | 0.268 | 0.183 | 0.346 | 0.207 | 0.375 | 0.462 | 0.586 | **0.097** | 0.253 |
| | 336 | 0.112 | **0.258** | 0.160 | 0.316 | 0.192 | 0.357 | 0.144 | 0.297 | 0.222 | 0.387 | 0.230 | 0.398 | 0.422 | 0.564 | **0.109** | 0.263 |
| | 720 | 0.148 | 0.306 | 0.179 | 0.345 | 0.235 | 0.408 | 0.183 | 0.347 | 0.269 | 0.435 | 0.273 | 0.463 | 0.438 | 0.578 | **0.142** | **0.302** |
| ETTh2 | 24 | **0.079** | 0.207 | 0.091 | 0.230 | 0.097 | 0.238 | 0.095 | 0.234 | 0.093 | 0.240 | 0.102 | 0.255 | 0.109 | 0.251 | 0.176 | 0.331 |
| | 48 | **0.118** | **0.259** | 0.124 | 0.274 | 0.131 | 0.281 | 0.130 | 0.279 | 0.155 | 0.314 | 0.169 | 0.348 | 0.147 | 0.302 | 0.202 | 0.357 |
| | 168 | **0.189** | **0.339** | 0.198 | 0.355 | 0.197 | 0.354 | 0.204 | 0.360 | 0.232 | 0.389 | 0.246 | 0.422 | 0.209 | 0.366 | 0.273 | 0.420 |
| | 336 | 0.206 | **0.360** | **0.205** | 0.364 | 0.207 | 0.366 | 0.206 | 0.364 | 0.263 | 0.417 | 0.267 | 0.437 | 0.237 | 0.391 | 0.284 | 0.423 |
| | 720 | 0.214 | 0.371 | 0.208 | 0.371 | 0.207 | 0.370 | 0.206 | 0.369 | 0.277 | 0.431 | 0.303 | 0.493 | **0.200** | **0.367** | 0.339 | 0.466 |
| ETTm1 | 24 | **0.015** | **0.088** | 0.016 | 0.093 | 0.019 | 0.103 | 0.015 | 0.091 | 0.030 | 0.137 | 0.065 | 0.202 | 0.027 | 0.127 | 0.027 | 0.128 |
| | 48 | **0.025** | **0.117** | 0.028 | 0.126 | 0.036 | 0.142 | 0.027 | 0.122 | 0.069 | 0.203 | 0.078 | 0.220 | 0.040 | 0.154 | 0.043 | 0.159 |
| | 96 | **0.038** | **0.147** | 0.045 | 0.162 | 0.054 | 0.178 | 0.041 | 0.153 | 0.194 | 0.372 | 0.199 | 0.386 | 0.097 | 0.246 | 0.054 | 0.178 |
| | 288 | **0.077** | **0.209** | 0.095 | 0.235 | 0.098 | 0.244 | 0.083 | 0.219 | 0.401 | 0.554 | 0.411 | 0.572 | 0.305 | 0.455 | 0.098 | 0.245 |
| | 672 | **0.113** | **0.257** | 0.142 | 0.290 | 0.136 | 0.290 | 0.122 | 0.268 | 0.512 | 0.644 | 0.598 | 0.702 | 0.445 | 0.576 | 0.121 | 0.274 |
| Electricity | 24 | **0.243** | **0.264** | 0.260 | 0.288 | 0.252 | 0.278 | 0.254 | 0.280 | 0.251 | 0.275 | 0.528 | 0.447 | **0.243** | 0.367 | - | - |
| | 48 | 0.292 | **0.300** | 0.313 | 0.321 | 0.300 | 0.308 | 0.304 | 0.314 | 0.346 | 0.339 | 0.409 | 0.414 | **0.283** | 0.397 | - | - |
| | 168 | 0.405 | **0.375** | 0.429 | 0.392 | 0.412 | 0.384 | 0.416 | 0.391 | 0.544 | 0.424 | 0.959 | 0.612 | **0.357** | 0.449 | - | - |
| | 336 | 0.560 | 0.473 | 0.565 | 0.478 | 0.548 | 0.466 | 0.556 | 0.482 | 0.713 | 0.512 | 1.079 | 0.639 | **0.355** | **0.446** | - | - |
| | 720 | 0.889 | 0.645 | 0.863 | 0.651 | 0.859 | 0.651 | 0.858 | 0.653 | 1.182 | 0.806 | 1.001 | 0.714 | **0.387** | **0.477** | - | - |
| Weather | 24 | **0.096** | **0.213** | **0.096** | 0.215 | 0.102 | 0.221 | 0.097 | 0.216 | 0.117 | 0.251 | 0.136 | 0.279 | 0.109 | 0.217 | 0.192 | 0.330 |
| | 48 | **0.138** | **0.262** | 0.140 | 0.264 | 0.139 | 0.264 | 0.140 | 0.264 | 0.178 | 0.318 | 0.206 | 0.356 | 0.143 | 0.269 | 0.231 | 0.361 |
| | 168 | 0.207 | 0.334 | 0.207 | 0.335 | 0.198 | 0.328 | 0.198 | 0.326 | 0.266 | 0.398 | 0.309 | 0.439 | **0.188** | **0.319** | 0.298 | 0.415 |
| | 336 | 0.230 | 0.356 | 0.231 | 0.360 | 0.215 | 0.347 | 0.220 | 0.350 | 0.297 | 0.416 | 0.359 | 0.484 | **0.192** | **0.320** | 0.314 | 0.429 |
| | 720 | 0.242 | 0.370 | 0.233 | 0.365 | 0.219 | 0.353 | 0.224 | 0.357 | 0.359 | 0.466 | 0.388 | 0.499 | **0.198** | **0.329** | 0.423 | 0.499 |
| Avg. | | **0.193** | **0.283** | 0.203 | 0.298 | 0.207 | 0.307 | 0.198 | 0.294 | 0.296 | 0.386 | 0.352 | 0.430 | 0.236 | 0.367 | - | - |

# H  RESULTS ON FEATURE-BASED FORECASTING BASELINES

Table 8: Multivariate forecasting results for feature-based approaches.

| Methods | | Representation Learning | | | | | | | | | | | | | | | Feature Engineered | |
| | | CoST | | TS2Vec | | TNC | | MoCo | | Triplet | | CPC | | TST | | TCC | | TSFresh | |
| Metrics | | MSE | MAE | MSE | MAE | MSE | MAE | MSE | MAE | MSE | MAE | MSE | MAE | MSE | MAE | MSE | MAE | MSE | MAE |
|---|---|---|---|---|---|---|---|---|---|---|---|---|---|---|---|---|---|---|---|
| ETTh1 | 24 | **0.386** | **0.429** | 0.590 | 0.531 | 0.708 | 0.592 | 0.623 | 0.555 | 0.942 | 0.729 | 0.728 | 0.600 | 0.735 | 0.633 | 0.766 | 0.629 | 3.858 | 1.574 |
| | 48 | **0.437** | **0.464** | 0.624 | 0.555 | 0.749 | 0.619 | 0.669 | 0.586 | 0.975 | 0.746 | 0.774 | 0.629 | 0.800 | 0.671 | 0.825 | 0.657 | 4.246 | 1.674 |
| | 168 | **0.643** | **0.582** | 0.762 | 0.639 | 0.884 | 0.699 | 0.820 | 0.674 | 1.135 | 0.825 | 0.920 | 0.714 | 0.973 | 0.768 | 0.982 | 0.731 | 3.527 | 1.500 |
| | 336 | **0.812** | **0.679** | 0.931 | 0.728 | 1.020 | 0.768 | 0.981 | 0.755 | 1.187 | 0.859 | 1.050 | 0.779 | 1.029 | 0.797 | 1.099 | 0.786 | 2.905 | 1.329 |
| | 720 | **0.970** | **0.771** | 1.063 | 0.799 | 1.157 | 0.830 | 1.138 | 0.831 | 1.283 | 0.916 | 1.160 | 0.835 | 1.020 | 0.798 | 1.267 | 0.859 | 2.667 | 1.283 |
| ETTh2 | 24 | 0.447 | 0.502 | **0.423** | **0.489** | 0.612 | 0.595 | 0.444 | 0.495 | 1.285 | 0.911 | 0.551 | 0.572 | 0.994 | 0.779 | 1.154 | 0.838 | 8.720 | 2.311 |
| | 48 | 0.699 | 0.637 | **0.619** | **0.605** | 0.840 | 0.716 | 0.613 | 0.595 | 1.455 | 0.966 | 0.752 | 0.684 | 1.159 | 0.850 | 1.579 | 0.983 | 12.771 | 2.746 |
| | 168 | **1.549** | **0.982** | 1.845 | 1.074 | 2.359 | 1.213 | 1.791 | 1.034 | 2.175 | 1.155 | 2.452 | 1.213 | 2.609 | 1.265 | 3.456 | 1.459 | 20.843 | 3.779 |
| | 336 | **1.749** | **1.042** | 2.194 | 1.197 | 2.782 | 1.349 | 2.241 | 1.186 | 2.007 | 1.101 | 2.664 | 1.304 | 2.824 | 1.337 | 3.184 | 1.420 | 14.801 | 3.006 |
| | 720 | **1.971** | **1.092** | 2.636 | 1.370 | 2.753 | 1.394 | 2.425 | 1.292 | 2.157 | 1.139 | 2.863 | 1.399 | 2.684 | 1.334 | 3.538 | 1.523 | 17.967 | 3.335 |
| ETTm1 | 24 | **0.246** | **0.329** | 0.453 | 0.444 | 0.522 | 0.472 | 0.458 | 0.444 | 0.689 | 0.592 | 0.478 | 0.459 | 0.471 | 0.491 | 0.502 | 0.478 | 0.639 | 0.589 |
| | 48 | **0.331** | **0.386** | 0.592 | 0.521 | 0.695 | 0.567 | 0.594 | 0.528 | 0.752 | 0.624 | 0.641 | 0.550 | 0.614 | 0.560 | 0.645 | 0.559 | 0.705 | 0.629 |
| | 96 | **0.378** | **0.419** | 0.635 | 0.554 | 0.731 | 0.595 | 0.621 | 0.553 | 0.744 | 0.623 | 0.707 | 0.593 | 0.645 | 0.581 | 0.675 | 0.583 | 0.675 | 0.606 |
| | 288 | **0.472** | **0.486** | 0.693 | 0.597 | 0.818 | 0.649 | 0.700 | 0.606 | 0.808 | 0.662 | 0.781 | 0.644 | 0.749 | 0.644 | 0.758 | 0.633 | 0.848 | 0.702 |
| | 672 | **0.620** | **0.574** | 0.782 | 0.653 | 0.932 | 0.712 | 0.821 | 0.674 | 0.917 | 0.720 | 0.880 | 0.700 | 0.857 | 0.709 | 0.854 | 0.689 | 0.968 | 0.767 |
| Electricity | 24 | **0.136** | **0.242** | 0.287 | 0.375 | 0.354 | 0.423 | 0.288 | 0.374 | 0.564 | 0.578 | 0.403 | 0.459 | 0.311 | 0.396 | 0.345 | 0.425 | - | - |
| | 48 | **0.153** | **0.258** | 0.309 | 0.391 | 0.376 | 0.438 | 0.310 | 0.390 | 0.569 | 0.581 | 0.424 | 0.473 | 0.326 | 0.407 | 0.365 | 0.439 | - | - |
| | 168 | **0.175** | **0.275** | 0.335 | 0.410 | 0.402 | 0.456 | 0.337 | 0.410 | 0.576 | 0.584 | 0.450 | 0.491 | 0.344 | 0.420 | 0.389 | 0.456 | - | - |
| | 336 | **0.196** | **0.296** | 0.351 | 0.422 | 0.417 | 0.466 | 0.353 | 0.422 | 0.591 | 0.591 | 0.466 | 0.501 | 0.359 | 0.431 | 0.407 | 0.468 | - | - |
| | 720 | **0.232** | **0.327** | 0.378 | 0.440 | 0.442 | 0.483 | 0.380 | 0.441 | 0.603 | 0.598 | 0.559 | 0.555 | 0.383 | 0.446 | 0.438 | 0.487 | - | - |
| Weather | 24 | **0.298** | **0.360** | 0.307 | 0.363 | 0.320 | 0.373 | 0.311 | 0.365 | 0.522 | 0.533 | 0.328 | 0.383 | 0.372 | 0.404 | 0.332 | 0.392 | 2.170 | 0.909 |
| | 48 | **0.359** | **0.411** | 0.374 | 0.418 | 0.380 | 0.421 | 0.372 | 0.416 | 0.539 | 0.543 | 0.390 | 0.433 | 0.418 | 0.445 | 0.391 | 0.439 | 2.235 | 0.936 |
| | 168 | **0.464** | **0.491** | 0.491 | 0.506 | 0.479 | 0.495 | 0.482 | 0.499 | 0.572 | 0.565 | 0.499 | 0.512 | 0.521 | 0.518 | 0.492 | 0.510 | 2.514 | 0.985 |
| | 336 | **0.497** | 0.517 | 0.525 | 0.530 | 0.505 | **0.514** | 0.516 | 0.523 | 0.582 | 0.572 | 0.533 | 0.536 | 0.555 | 0.541 | 0.523 | 0.532 | 2.293 | 0.969 |
| | 720 | 0.533 | 0.542 | 0.556 | 0.552 | **0.519** | **0.525** | 0.540 | 0.540 | 0.597 | 0.582 | 0.559 | 0.553 | 0.575 | 0.555 | 0.548 | 0.549 | 2.468 | 0.961 |
| Avg. | | **0.590** | **0.524** | 0.750 | 0.607 | 0.870 | 0.655 | 0.753 | 0.608 | 0.969 | 0.732 | 0.880 | 0.663 | 0.893 | 0.671 | 1.021 | 0.701 | - | - |

Table 9: Univariate forecasting results for feature-based approaches

| Methods | | Representation Learning | | | | | | | | | | | | | | | Feature Engineered | |
| | | CoST | | TS2Vec | | TNC | | MoCo | | Triplet | | CPC | | TST | | TCC | | TSFresh | |
| Metrics | | MSE | MAE | MSE | MAE | MSE | MAE | MSE | MAE | MSE | MAE | MSE | MAE | MSE | MAE | MSE | MAE | MSE | MAE |
|---|---|---|---|---|---|---|---|---|---|---|---|---|---|---|---|---|---|---|---|
| ETTh1 | 24 | 0.040 | 0.152 | **0.039** | **0.151** | 0.057 | 0.184 | 0.040 | **0.151** | 0.130 | 0.289 | 0.076 | 0.217 | 0.127 | 0.284 | 0.053 | 0.175 | 0.080 | 0.224 |
| | 48 | **0.060** | **0.186** | 0.062 | 0.189 | 0.094 | 0.239 | 0.063 | 0.191 | 0.145 | 0.306 | 0.104 | 0.259 | 0.202 | 0.362 | 0.074 | 0.209 | 0.092 | 0.242 |
| | 168 | **0.097** | **0.236** | 0.142 | 0.291 | 0.171 | 0.329 | 0.122 | 0.268 | 0.173 | 0.336 | 0.162 | 0.326 | 0.491 | 0.596 | 0.133 | 0.284 | **0.097** | 0.253 |
| | 336 | 0.112 | **0.258** | 0.160 | 0.316 | 0.192 | 0.357 | 0.144 | 0.297 | 0.167 | 0.333 | 0.183 | 0.351 | 0.526 | 0.618 | 0.161 | 0.320 | **0.109** | 0.263 |
| | 720 | 0.148 | 0.306 | 0.179 | 0.345 | 0.235 | 0.408 | 0.183 | 0.347 | 0.195 | 0.368 | 0.212 | 0.387 | 0.717 | 0.760 | 0.176 | 0.343 | **0.142** | **0.302** |
| ETTh2 | 24 | **0.079** | **0.207** | 0.091 | 0.230 | 0.097 | 0.238 | 0.095 | 0.234 | 0.160 | 0.316 | 0.109 | 0.251 | 0.134 | 0.281 | 0.111 | 0.255 | 0.176 | 0.331 |
| | 48 | **0.118** | **0.259** | 0.124 | 0.274 | 0.131 | 0.281 | 0.130 | 0.279 | 0.181 | 0.339 | 0.152 | 0.301 | 0.171 | 0.321 | 0.148 | 0.298 | 0.202 | 0.357 |
| | 168 | **0.189** | **0.339** | 0.198 | 0.355 | 0.197 | 0.354 | 0.204 | 0.360 | 0.214 | 0.372 | 0.251 | 0.392 | 0.261 | 0.404 | 0.225 | 0.374 | 0.273 | 0.420 |
| | 336 | 0.206 | **0.360** | **0.205** | 0.364 | 0.207 | 0.366 | 0.206 | 0.364 | 0.232 | 0.389 | 0.238 | 0.388 | 0.269 | 0.413 | 0.232 | 0.385 | 0.284 | 0.423 |
| | 720 | 0.214 | 0.371 | 0.208 | 0.371 | 0.207 | 0.370 | **0.206** | **0.369** | 0.251 | 0.406 | 0.234 | 0.389 | 0.278 | 0.420 | 0.242 | 0.397 | 0.339 | 0.466 |
| ETTm1 | 24 | **0.015** | **0.088** | 0.016 | 0.093 | 0.019 | 0.103 | **0.015** | 0.091 | 0.071 | 0.180 | 0.018 | 0.102 | 0.048 | 0.151 | 0.026 | 0.122 | 0.027 | 0.128 |
| | 48 | **0.025** | **0.117** | 0.028 | 0.126 | 0.036 | 0.142 | 0.027 | 0.122 | 0.084 | 0.206 | 0.035 | 0.142 | 0.064 | 0.183 | 0.045 | 0.165 | 0.043 | 0.159 |
| | 96 | **0.038** | **0.147** | 0.045 | 0.162 | 0.054 | 0.178 | 0.041 | 0.153 | 0.097 | 0.230 | 0.059 | 0.188 | 0.102 | 0.231 | 0.072 | 0.211 | 0.054 | 0.178 |
| | 288 | **0.077** | **0.209** | 0.095 | 0.235 | 0.098 | 0.244 | 0.083 | 0.219 | 0.130 | 0.276 | 0.118 | 0.271 | 0.172 | 0.316 | 0.158 | 0.318 | 0.098 | 0.245 |
| | 672 | **0.113** | **0.257** | 0.142 | 0.290 | 0.136 | 0.290 | 0.122 | 0.268 | 0.160 | 0.315 | 0.177 | 0.332 | 0.224 | 0.366 | 0.239 | 0.398 | 0.121 | 0.274 |
| Electricity | 24 | **0.243** | **0.264** | 0.260 | 0.288 | 0.252 | 0.278 | 0.254 | 0.280 | 0.355 | 0.379 | 0.264 | 0.299 | 0.351 | 0.387 | 0.266 | 0.301 | - | - |
| | 48 | **0.292** | **0.300** | 0.313 | 0.321 | 0.300 | 0.308 | 0.304 | 0.314 | 0.375 | 0.390 | 0.321 | 0.339 | 0.398 | 0.416 | 0.317 | 0.330 | - | - |
| | 168 | **0.405** | **0.375** | 0.429 | 0.392 | 0.412 | 0.384 | 0.416 | 0.391 | 0.482 | 0.459 | 0.438 | 0.418 | 0.535 | 0.498 | 0.424 | 0.402 | - | - |
| | 336 | 0.560 | 0.473 | 0.565 | 0.478 | **0.548** | **0.466** | 0.556 | 0.482 | 0.633 | 0.551 | 0.599 | 0.507 | 0.656 | 0.575 | 0.578 | 0.486 | - | - |
| | 720 | 0.889 | **0.645** | 0.863 | 0.651 | 0.859 | 0.651 | **0.858** | 0.653 | 0.930 | 0.706 | 0.957 | 0.679 | 0.929 | 0.729 | 0.950 | 0.667 | - | - |
| Weather | 24 | **0.096** | **0.213** | 0.096 | 0.215 | 0.102 | 0.221 | 0.097 | 0.216 | 0.203 | 0.337 | 0.105 | 0.226 | 0.124 | 0.244 | 0.107 | 0.232 | 0.192 | 0.330 |
| | 48 | **0.138** | **0.262** | 0.140 | 0.264 | 0.139 | 0.264 | 0.140 | 0.264 | 0.219 | 0.351 | 0.147 | 0.272 | 0.151 | 0.280 | 0.143 | 0.272 | 0.231 | 0.361 |
| | 168 | 0.207 | 0.334 | 0.207 | 0.335 | **0.198** | 0.328 | **0.198** | **0.326** | 0.251 | 0.379 | 0.213 | 0.340 | 0.213 | 0.342 | 0.204 | 0.333 | 0.298 | 0.415 |
| | 336 | 0.230 | 0.356 | 0.231 | 0.360 | **0.215** | **0.347** | 0.220 | 0.350 | 0.262 | 0.389 | 0.234 | 0.362 | 0.233 | 0.361 | 0.219 | 0.350 | 0.314 | 0.429 |
| | 720 | 0.242 | 0.370 | 0.233 | 0.365 | **0.219** | **0.353** | 0.224 | 0.357 | 0.263 | 0.394 | 0.237 | 0.366 | 0.232 | 0.361 | 0.220 | 0.352 | 0.423 | 0.499 |
| Avg. | | **0.193** | **0.283** | 0.203 | 0.298 | 0.207 | 0.307 | 0.198 | 0.294 | 0.255 | 0.360 | 0.226 | 0.324 | 0.304 | 0.396 | 0.221 | 0.319 | - | - |

We include hand-crafted features (using the same experiment methodology as representation learning approaches) in our benchmark, by using features from the TSFresh package. We select the same set of features for all datasets and settings, to avoid extensive feature engineering which requires domain expertise. TSFresh generally under performs in the multivariate benchmark due to the high dimensionality of the generated features, since it extracts univariate features and thus feature size increases linearly with input size. On the other hand, it performs relatively well on the univariate setting.

# I  RESULTS ON END-TO-END FORECASTING BASELINES COMPARED TO COST

Table 10: Multivariate forecasting results for End-to-end forecasting baselines compared to CoST

| Methods | | End-to-end Forecasting | | | | | | | | | |
| | | CoST | | Informer | | LogTrans | | TCN | | LSTnet | |
| Metrics | | MSE | MAE | MSE | MAE | MSE | MAE | MSE | MAE | MSE | MAE |
|---|---|---|---|---|---|---|---|---|---|---|---|
| ETTh1 | 24 | **0.386** | **0.429** | 0.577 | 0.549 | 0.686 | 0.604 | 0.583 | 0.547 | 1.293 | 0.901 |
| | 48 | **0.437** | **0.464** | 0.685 | 0.625 | 0.766 | 0.757 | 0.670 | 0.606 | 1.456 | 0.960 |
| | 168 | **0.643** | **0.582** | 0.931 | 0.752 | 1.002 | 0.846 | 0.811 | 0.680 | 1.997 | 1.214 |
| | 336 | **0.812** | **0.679** | 1.128 | 0.873 | 1.362 | 0.952 | 1.132 | 0.815 | 2.655 | 1.369 |
| | 720 | **0.970** | **0.771** | 1.215 | 0.896 | 1.397 | 1.291 | 1.165 | 0.813 | 2.143 | 1.380 |
| ETTh2 | 24 | **0.447** | **0.502** | 0.720 | 0.665 | 0.828 | 0.750 | 0.935 | 0.754 | 2.742 | 1.457 |
| | 48 | **0.699** | **0.637** | 1.457 | 1.001 | 1.806 | 1.034 | 1.300 | 0.911 | 3.567 | 1.687 |
| | 168 | **1.549** | **0.982** | 3.489 | 1.515 | 4.070 | 1.681 | 4.017 | 1.579 | 3.242 | 2.513 |
| | 336 | **1.749** | **1.042** | 2.723 | 1.340 | 3.875 | 1.763 | 3.460 | 1.456 | 2.544 | 2.591 |
| | 720 | **1.971** | **1.092** | 3.467 | 1.473 | 3.913 | 1.552 | 3.106 | 1.381 | 4.625 | 3.709 |
| ETTm1 | 24 | **0.246** | **0.329** | 0.323 | 0.369 | 0.419 | 0.412 | 0.363 | 0.397 | 1.968 | 1.170 |
| | 48 | **0.331** | **0.386** | 0.494 | 0.503 | 0.507 | 0.583 | 0.542 | 0.508 | 1.999 | 1.215 |
| | 96 | **0.378** | **0.419** | 0.678 | 0.614 | 0.768 | 0.792 | 0.666 | 0.578 | 2.762 | 1.542 |
| | 288 | **0.472** | **0.486** | 1.056 | 0.786 | 1.462 | 1.320 | 0.991 | 0.735 | 1.257 | 2.076 |
| | 672 | **0.620** | **0.574** | 1.192 | 0.926 | 1.669 | 1.461 | 1.032 | 0.756 | 1.917 | 2.941 |
| Electricity | 24 | **0.136** | **0.242** | 0.312 | 0.387 | 0.297 | 0.374 | 0.235 | 0.346 | 0.356 | 0.419 |
| | 48 | **0.153** | **0.258** | 0.392 | 0.431 | 0.316 | 0.389 | 0.253 | 0.359 | 0.429 | 0.456 |
| | 168 | **0.175** | **0.275** | 0.515 | 0.509 | 0.426 | 0.466 | 0.278 | 0.372 | 0.372 | 0.425 |
| | 336 | **0.196** | **0.296** | 0.759 | 0.625 | 0.365 | 0.417 | 0.287 | 0.382 | 0.352 | 0.409 |
| | 720 | **0.232** | **0.327** | 0.969 | 0.788 | 0.344 | 0.403 | 0.287 | 0.381 | 0.38 | 0.443 |
| Weather | 24 | **0.298** | **0.360** | 0.335 | 0.381 | 0.435 | 0.477 | 0.321 | 0.367 | 0.615 | 0.545 |
| | 48 | **0.359** | **0.411** | 0.395 | 0.459 | 0.426 | 0.495 | 0.386 | 0.423 | 0.66 | 0.589 |
| | 168 | **0.464** | **0.491** | 0.608 | 0.567 | 0.727 | 0.671 | 0.491 | 0.501 | 0.748 | 0.647 |
| | 336 | **0.497** | 0.517 | 0.702 | 0.620 | 0.754 | 0.670 | 0.502 | **0.507** | 0.782 | 0.683 |
| | 720 | 0.533 | 0.542 | 0.831 | 0.731 | 0.885 | 0.773 | **0.498** | **0.508** | 0.851 | 0.757 |
| Avg. | | **0.590** | **0.524** | 1.038 | 0.735 | 1.180 | 0.837 | 0.972 | 0.666 | 1.668 | 1.284 |

Table 11: Univariate forecasting results for end-to-end forecasting baselines compared to CoST

| Methods | | End-to-end Forecasting | | | | | | | | | |
| | | CoST | | Informer | | LogTrans | | TCN | | LSTnet | |
| Metrics | | MSE | MAE | MSE | MAE | MSE | MAE | MSE | MAE | MSE | MAE |
|---|---|---|---|---|---|---|---|---|---|---|---|
| ETTh1 | 24 | 0.040 | 0.152 | 0.098 | 0.247 | 0.103 | 0.259 | 0.104 | 0.254 | 0.108 | 0.284 |
| | 48 | **0.060** | **0.186** | 0.158 | 0.319 | 0.167 | 0.328 | 0.206 | 0.366 | 0.175 | 0.424 |
| | 168 | **0.097** | **0.236** | 0.183 | 0.346 | 0.207 | 0.375 | 0.462 | 0.586 | 0.396 | 0.504 |
| | 336 | 0.112 | **0.258** | 0.222 | 0.387 | 0.230 | 0.398 | 0.422 | 0.564 | 0.468 | 0.593 |
| | 720 | 0.148 | 0.306 | 0.269 | 0.435 | 0.273 | 0.463 | 0.438 | 0.578 | 0.659 | 0.766 |
| ETTh2 | 24 | **0.079** | 0.207 | 0.093 | 0.240 | 0.102 | 0.255 | 0.109 | 0.251 | 3.554 | 0.445 |
| | 48 | **0.118** | **0.259** | 0.155 | 0.314 | 0.169 | 0.348 | 0.147 | 0.302 | 3.190 | 0.474 |
| | 168 | **0.189** | **0.339** | 0.232 | 0.389 | 0.246 | 0.422 | 0.209 | 0.366 | 2.800 | 0.595 |
| | 336 | 0.206 | **0.360** | 0.263 | 0.417 | 0.267 | 0.437 | 0.237 | 0.391 | 2.753 | 0.738 |
| | 720 | 0.214 | 0.371 | 0.277 | 0.431 | 0.303 | 0.493 | 0.200 | 0.367 | 2.878 | 1.044 |
| ETTm1 | 24 | **0.015** | **0.088** | 0.030 | 0.137 | 0.065 | 0.202 | 0.027 | 0.127 | 0.090 | 0.206 |
| | 48 | **0.025** | **0.117** | 0.069 | 0.203 | 0.078 | 0.220 | 0.040 | 0.154 | 0.179 | 0.306 |
| | 96 | **0.038** | **0.147** | 0.194 | 0.372 | 0.199 | 0.386 | 0.097 | 0.246 | 0.272 | 0.399 |
| | 288 | **0.077** | **0.209** | 0.401 | 0.554 | 0.411 | 0.572 | 0.305 | 0.455 | 0.462 | 0.558 |
| | 672 | **0.113** | **0.257** | 0.512 | 0.644 | 0.598 | 0.702 | 0.445 | 0.576 | 0.639 | 0.697 |
| Electricity | 24 | **0.243** | **0.264** | 0.251 | 0.275 | 0.528 | 0.447 | **0.243** | 0.367 | 0.281 | 0.287 |
| | 48 | 0.292 | **0.300** | 0.346 | 0.339 | 0.409 | 0.414 | **0.283** | 0.397 | 0.381 | 0.366 |
| | 168 | 0.405 | 0.375 | 0.544 | 0.424 | 0.959 | 0.612 | **0.357** | **0.449** | 0.599 | 0.500 |
| | 336 | 0.560 | 0.473 | 0.713 | 0.512 | 1.079 | 0.639 | **0.355** | **0.446** | 0.823 | 0.624 |
| | 720 | 0.889 | 0.645 | 1.182 | 0.806 | 1.001 | 0.714 | **0.387** | **0.477** | 1.278 | 0.906 |
| Weather | 24 | **0.096** | **0.213** | 0.117 | 0.251 | 0.136 | 0.279 | 0.109 | 0.217 | - | - |
| | 48 | **0.138** | **0.262** | 0.178 | 0.318 | 0.206 | 0.356 | 0.143 | 0.269 | - | - |
| | 168 | 0.207 | 0.334 | 0.266 | 0.398 | 0.309 | 0.439 | **0.188** | **0.319** | - | - |
| | 336 | 0.230 | 0.356 | 0.297 | 0.416 | 0.359 | 0.484 | **0.192** | **0.320** | - | - |
| | 720 | 0.242 | 0.370 | 0.359 | 0.466 | 0.388 | 0.499 | **0.198** | **0.329** | - | - |
| Avg. | | **0.193** | **0.283** | 0.296 | 0.386 | 0.352 | 0.430 | 0.236 | 0.367 | - | - |

## J  RESULTS ON M5 DATASETS

Table 12: Results on M5 datasets (Makridakis et al., 2020). M5 is a multivariate dataset, with forecast horizon is 28, as per M5 competition settings.

| Methods | | | | | Representation Learning | | | | | End-to-end Forecasting | | |
| --- | --- | --- | --- | --- | --- | --- | --- | --- | --- | --- | --- | --- |
| | CoST | | TS2Vec | | TNC | | MoCo | | Informer | | TCN | |
| Metrics | MSE | MAE | MSE | MAE | MSE | MAE | MSE | MAE | MSE | MAE | MSE | MAE |
| L1 | **0.063** | **0.211** | 0.299 | 0.446 | 0.671 | 0.627 | 0.279 | 0.415 | 0.836 | 0.724 | 1.395 | 1.020 |
| L2 | **0.154** | **0.311** | 0.383 | 0.498 | 0.521 | 0.564 | 0.360 | 0.482 | 1.436 | 0.991 | 1.310 | 0.932 |
| L3 | **0.191** | **0.340** | 0.591 | 0.549 | 0.621 | 0.582 | 0.438 | 0.496 | 1.747 | 1.033 | 1.847 | 1.048 |
| L4 | **0.149** | **0.293** | 0.291 | 0.403 | 0.393 | 0.482 | 0.304 | 0.408 | 1.023 | 0.829 | 1.388 | 0.992 |
| L5 | **0.260** | **0.390** | 0.419 | 0.484 | 0.506 | 0.545 | 0.462 | 0.508 | 1.514 | 0.899 | 2.039 | 1.172 |
| L6 | **0.255** | **0.386** | 0.500 | 0.528 | 0.586 | 0.589 | 0.478 | 0.510 | 1.099 | 0.822 | 1.309 | 0.925 |
| L7 | **0.394** | **0.482** | 0.630 | 0.600 | 0.648 | 0.618 | 0.641 | 0.605 | 1.565 | 0.931 | 1.475 | 0.953 |
| L8 | **0.328** | **0.442** | 0.589 | 0.559 | 0.614 | 0.584 | 0.610 | 0.575 | 1.969 | 1.063 | 1.691 | 0.990 |
| L9 | **0.702** | **0.582** | 2.150 | 0.748 | 1.445 | 0.726 | 1.293 | 0.701 | 4.923 | 1.313 | 4.634 | 1.332 |
| L10 | **1.471** | **0.783** | 1.677 | 0.812 | 1.679 | 0.830 | 1.604 | 0.815 | 2.474 | 0.977 | 2.476 | 1.025 |
| Avg. | **0.397** | **0.422** | 0.753 | 0.563 | 0.769 | 0.615 | 0.647 | 0.551 | 1.859 | 0.958 | 1.957 | 1.039 |

## K  CASE STUDY: DISENTANGLEMENT

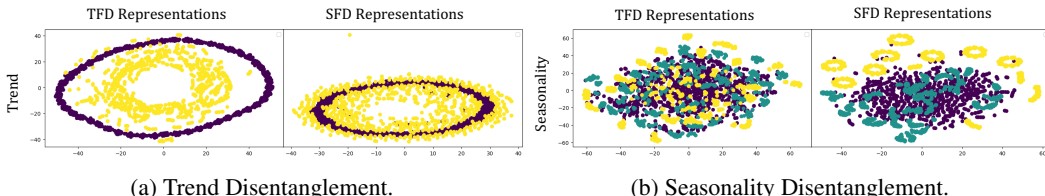

(a) Trend Disentanglement.    (b) Seasonality Disentanglement.

Figure 5: T-SNE visualization of seasonal-trend disentanglement in CoST embeddings. TFD Representations refer to the representations generated by the Trend Feature Disentangler while SFD Representations refer to the representations generated by the Seasonal Feature Disentangler. (a) We select a single seasonality and visualize the representations. The two colors represent the two distinct trends. (b) We select a single trend and visualize the representations. The three colors represent the three distinct seasonal patterns.

To exhibit CoST's ability to disentangle trend and seasonal components, we plot the T-SNE representations of the Trend Feature Disentangler (TFD) and Seasonal Feature Disentangler (SFD) separately. From Figure 5a, we see that representations from the TFD indeed have better clusterability and the representations from SFD have a degree of overlap between the two trend patterns. From Figure 5b, we see that the TFD representations have a higher degree of overlap between the three seasonality patterns than the SFD representations. Here, to better highlight the stronger capability of seasonality representations in extracting seasonal patterns, we used complex seasonality patterns (described below).

**Complex Seasonality**    Similar to the synthetic data generation process in Appendix D, we generate three different seasonality patterns before combining them with the two trend patterns. The first seasonality pattern is no seasonality. The second pattern begins with a sine wave of period, phase, and amplitude of $\{3, 0, 10\}$, thereafter, a mask is then applied to the entire pattern, consisting of a repeating pattern of three 1s and seven 0s. The third pattern begins with a sine wave of period, phase, and amplitude of $\{10, 0.5, 15\}$, thereafter, a mask is then applied to the entire pattern, consisting of a repeating pattern of two 1s and eight 0s.

