# OpenReview forum: "CoST: Contrastive Learning of Disentangled Seasonal-Trend Representations for Time Series Forecasting"
_ICLR.cc/2022/Conference — ICLR 2022 Poster_

### Official Review · Reviewer_bLmw · 2021-11-01

**Correctness:** 3
**Technical Novelty And Significance:** 3
**Empirical Novelty And Significance:** 3
**Recommendation:** 5
**Confidence:** 4

**Main Review:**

I enjoyed reading the paper. The method for representation learning is original and clearly described and there's not much for me to argue around.

My main criticism comes from the experiments and the discussion/awareness of related work. I think that these criticisms can be taken care of by editing/toning down the paper and as part of the review process.

It's rather surprising that an end-to-end forecast approach can be beaten (by a margin) by a two-step approach. This needs much more and much stronger explanation and the one given here (because of the trend-seasonal-decomposition) does not strike me as strong enough -- a transformer-based model with enough parameters should learn this easily for example and there are plenty of other STL-based approaches in forecasting (e.g., implemented in the R Forecast package) that then should fare well.

In most of the literature on contrastive learning, the performance gets close to the state of the art but I'm not aware of it beating the state of the art by such a margin. Alas, it's unclear to me how the different representation learning approaches are used to produce forecasts. I may have overread this, but an explanation of how we go from representation to forecast would be needed for the main results (there is an ablation study with Table 5, but I'm not sure what was used for the main results in Table 1). In a typical contrastive learning paper, one would have a simple model (e.g., linear) on top of the complex representations and then get a similar performance to the state of the art. Why is a similar empirical effectiveness not happening here and instead, so much more improvement?

In the experiments, I'm missing natural baselines (e.g., NBEATS or TFT or DeepAR) and datasets that are a bit more standard, like the M5 data sets for example. Of course, reviewers always want more data sets and more methods, but given the claim that end-to-end SOTA approaches are beaten so convincingly, this leaves me wondering whether indeed SOTA methods were used in comparison or whether the data sets used are representative of the forecasting task.

Also, using the embeddings as co-variates for an end-to-end-learnt forecasting model would be a natural experiment to explain the accuracy gain better (that would basically be Table 5 with another regressor on top)

Further questions/comments:
- overfitting doesn't seem to be a major problem in forecasting where model sizes are comparatively small (when compared with NLP for example), so this motivation doesn't convince me (page 1). A more convincing angle may be greater generality or speed.

- page 2: "Firstly, we highlight that existing work using end-to-end deep forecasting methods directly model
the time-lagged relationship along the observed data X"  There is a considerable body of literature on probabilistic and deep forecasting models (e.g., https://iclr.cc/virtual/2021/spotlight/3409, but also Deep State Space Models at NeurIPS 20219 and much follow up work) which are in direct contradiction to this statement. In particular for multi-variate probabilistic and deep time series models, there is a growing body of literature which addresses the fact that Figure 2 is an over-simplification in the multi-variate context -- multiple X can have direct causal interaction (e.g., products that cannabilize/substitute/cross-sell)

- experiments: are the embeddings trained on a single data set or across data sets? Over which time horizons do we train the embeddings and how do we forecast (rolling window evaluations), etc? Why is MAE and MSE chosen when there are things like sMAPE and MASE which are more generally accepted?



**Summary Of The Paper:**

This paper proposes a novel way to represent time series learnt with contrastive losses in both the time and frequency domain for the forecasting task. The approach is novel and timely and is clearly described. The authors make a connection to causality which they use also for designing data augmentation schemes.

**Summary Of The Review:**

Overall, I'd like to recommend the paper for acceptance given the clear merits of its representation learning approach which I applaud the authors for. However, I have some open questions which I would like to clarify as part of the review process so right now, my evaluation is cautious.

---

> ### Author Response · Authors · 2021-11-17
> **Response to Reviewer bLmw (1/5)**
>
> Thank you for your detailed review and suggestions for improvements, we are glad that you enjoyed reading the paper.
>
> #Q1: why the two-stage approach is able to beat existing end-to-end approaches
>
> The reasons to why the two-stage approach is able to beat existing end-to-end approaches comes in two parts:
>
> 1. Contrastive learning is able to learn robust representations via interventions on the error variable, whereas end-to-end approaches tend to overfit on spurious correlations. In the domain of CV, contrastive learning has matched the performance of supervised learning methods, but still has not beaten them. However, when dealing with time series, the data is noisier and more non-stationary. Thus, the robust representations from contrastive learning brings about a larger improvement in performance than in CV.
>
> 2. Our seasonal-trend representations are a powerful inductive bias for time series forecasting. Our STL-based approach is different from existing forecasting approaches - we emphasize that our seasonal-trend representations lies in latent space, while traditional STL based methods do it in input space. CoST has a much stronger expressive capability to uncover complex seasonal and trend patterns. Furthermore, we argue that a large transformer-based model with sufficient parameters is unable to learn such patterns because time series data is very noisy and it is easy for a large model to learn spurious autocorrelations - incorporating the proper inductive bias is crucial to help the model generalize.
>
> #Q2: more baselines/datasets:
>
> We also perform further experiments on the requested baselines/experiments:
>
> N-BEATS & DeepAR:
>
> | Methods |     | CoST  |       | Informer |       | LogTrans |       | TCN   |       | LSTnet |       | N-BEATS |       | DeepAR |       |
> |---------|-----|-------|-------|----------|-------|----------|-------|-------|-------|--------|-------|---------|-------|--------|-------|
> | Metrics |     | MSE   | MAE   | MSE      | MAE   | MSE      | MAE   | MSE   | MAE   | MSE    | MAE   | MSE     | MAE   | MSE    | MAE   |
> | ETTh1   | 24  | 0.040 | 0.152 | 0.098    | 0.247 | 0.103    | 0.259 | 0.104 | 0.254 | 0.108  | 0.284 | 0.094   | 0.238 | 0.107  | 0.28  |
> |         | 48  | 0.060 | 0.186 | 0.158    | 0.319 | 0.167    | 0.328 | 0.206 | 0.366 | 0.175  | 0.424 | 0.210   | 0.367 | 0.162  | 0.327 |
> |         | 168 | 0.097 | 0.236 | 0.183    | 0.346 | 0.207    | 0.375 | 0.462 | 0.586 | 0.396  | 0.504 | 0.232   | 0.391 | 0.239  | 0.422 |
> |         | 336 | 0.112 | 0.258 | 0.222    | 0.387 | 0.230    | 0.398 | 0.422 | 0.564 | 0.468  | 0.593 | 0.232   | 0.388 | 0.445  | 0.552 |
> |         | 720 | 0.148 | 0.306 | 0.269    | 0.435 | 0.273    | 0.463 | 0.438 | 0.578 | 0.659  | 0.766 | 0.322   | 0.490 | 0.658  | 0.707 |
> | ETTh2   | 24  | 0.079 | 0.207 | 0.093    | 0.240 | 0.102    | 0.255 | 0.109 | 0.251 | 3.554  | 0.445 | 0.198   | 0.345 | 0.098  | 0.263 |
> |         | 48  | 0.118 | 0.259 | 0.155    | 0.314 | 0.169    | 0.348 | 0.147 | 0.302 | 3.190  | 0.474 | 0.234   | 0.386 | 0.163  | 0.341 |
> |         | 168 | 0.189 | 0.339 | 0.232    | 0.389 | 0.246    | 0.422 | 0.209 | 0.366 | 2.800  | 0.595 | 0.331   | 0.453 | 0.255  | 0.414 |
> |         | 336 | 0.206 | 0.360 | 0.263    | 0.417 | 0.267    | 0.437 | 0.237 | 0.391 | 2.753  | 0.738 | 0.431   | 0.508 | 0.604  | 0.607 |
> |         | 720 | 0.214 | 0.371 | 0.277    | 0.431 | 0.303    | 0.493 | 0.200 | 0.367 | 2.878  | 1.044 | 0.437   | 0.517 | 0.429  | 0.58  |
> | ETTm1   | 24  | 0.015 | 0.088 | 0.030    | 0.137 | 0.065    | 0.202 | 0.027 | 0.127 | 0.090  | 0.206 | 0.054   | 0.184 | 0.091  | 0.243 |
> |         | 48  | 0.025 | 0.117 | 0.069    | 0.203 | 0.078    | 0.220 | 0.040 | 0.154 | 0.179  | 0.306 | 0.190   | 0.361 | 0.219  | 0.362 |
> |         | 96  | 0.038 | 0.147 | 0.194    | 0.372 | 0.199    | 0.386 | 0.097 | 0.246 | 0.272  | 0.399 | 0.183   | 0.353 | 0.364  | 0.496 |
> |         | 288 | 0.077 | 0.209 | 0.401    | 0.554 | 0.411    | 0.572 | 0.305 | 0.455 | 0.462  | 0.558 | 0.186   | 0.362 | 0.948  | 0.795 |
> |         | 672 | 0.113 | 0.257 | 0.512    | 0.644 | 0.598    | 0.702 | 0.445 | 0.576 | 0.639  | 0.697 | 0.197   | 0.368 | 2.437  | 1.352 |
> | Avg.    |     | 0.102 | 0.233 | 0.210    | 0.362 | 0.228    | 0.391 | 0.230 | 0.372 | 1.242  | 0.536 | 0.235   | 0.381 | 0.481  | 0.516 |
>
> N-BEATS and DeepAR are univariate models - results presented are on the univariate benchmark. As can be seen, these baseline approaches underperform in the long time series forecasting setting.

---

> > ### Author Response · Authors · 2021-11-17
> > **Response to Reviewer bLmw (2/5)**
> >
> > M5 Dataset:
> >
> > | Methods |     | CoST  |       | TS2Vec |       | TNC   |       | MoCo  |       | Informer |       | TCN   |       |
> > |---------|-----|-------|-------|--------|-------|-------|-------|-------|-------|----------|-------|-------|-------|
> > | Metrics |     | MSE   | MAE   | MSE    | MAE   | MSE   | MAE   | MSE   | MAE   | MSE      | MAE   | MSE   | MAE   |
> > | M5      | L1  | 0.063 | 0.211 | 0.299  | 0.446 | 0.671 | 0.627 | 0.279 | 0.415 | 0.836    | 0.724 | 1.395 | 1.020 |
> > |         | L2  | 0.154 | 0.311 | 0.383  | 0.498 | 0.521 | 0.564 | 0.360 | 0.482 | 1.436    | 0.991 | 1.310 | 0.932 |
> > |         | L3  | 0.191 | 0.340 | 0.591  | 0.549 | 0.621 | 0.582 | 0.438 | 0.496 | 1.747    | 1.033 | 1.847 | 1.048 |
> > |         | L4  | 0.149 | 0.293 | 0.291  | 0.403 | 0.393 | 0.482 | 0.304 | 0.408 | 1.023    | 0.829 | 1.388 | 0.992 |
> > |         | L5  | 0.260 | 0.390 | 0.419  | 0.484 | 0.506 | 0.545 | 0.462 | 0.508 | 1.514    | 0.899 | 2.039 | 1.172 |
> > |         | L6  | 0.255 | 0.386 | 0.500  | 0.528 | 0.586 | 0.589 | 0.478 | 0.510 | 1.099    | 0.822 | 1.309 | 0.925 |
> > |         | L7  | 0.394 | 0.482 | 0.630  | 0.600 | 0.648 | 0.618 | 0.641 | 0.605 | 1.565    | 0.931 | 1.475 | 0.953 |
> > |         | L8  | 0.328 | 0.442 | 0.589  | 0.559 | 0.614 | 0.584 | 0.610 | 0.575 | 1.969    | 1.063 | 1.691 | 0.990 |
> > |         | L9  | 0.702 | 0.582 | 2.150  | 0.748 | 1.445 | 0.726 | 1.293 | 0.701 | 4.923    | 1.313 | 4.634 | 1.332 |
> > |         | L10 | 1.471 | 0.783 | 1.677  | 0.812 | 1.679 | 0.830 | 1.604 | 0.815 | 2.474    | 0.977 | 2.476 | 1.025 |
> > | Avg.    |     | 0.397 | 0.422 | 0.753  | 0.563 | 0.769 | 0.615 | 0.647 | 0.551 | 1.859    | 0.958 | 1.957 | 1.039 |
> >
> > We provide the results on the subsets of L1 - L10 of the M5 dataset, for several representation learning and end-to-end approaches. Overall, we can see that the ranking of the approaches are consistent with other datasets.
> >
> > #Q3: “Also, using the embeddings as co-variates for an end-to-end-learnt forecasting model would be a natural experiment to explain the accuracy gain better (that would basically be Table 5 with another regressor on top)“
> >
> > We have performed the experiments which uses CoST embeddings (frozen) as covariates for Informer. We see that even with CoST embeddings as covariates for the informer, the results are similar to that of vanilla Informer - this suggests that Informer does not have the proper inductive bias in the model architecture to produce good forecasts.
> >
> > | Methods  | CoST  |       | Informer |       | LogTrans |       | TCN   |       | LSTnet |       | Informer (Embedding Covariate) |       |
> > |----------|-------|-------|----------|-------|----------|-------|-------|-------|--------|-------|--------------------------------|-------|
> > | Metrics  | MSE   | MAE   | MSE      | MAE   | MSE      | MAE   | MSE   | MAE   | MSE    | MAE   | MSE                            | MAE   |
> > | Multivar | 0.781 | 0.625 | 1.342    | 0.859 | 1.635    | 1.053 | 1.385 | 0.834 | 2.411  | 1.782 | 1.323                          | 0.841 |
> > | Univar   | 0.102 | 0.233 | 0.210    | 0.362 | 0.228    | 0.391 | 0.230 | 0.372 | 1.242  | 0.536 | 0.243                          | 0.395 |
> >
> > For these additional baselines and experiments, we will be adding them into the final submission into the appropriate sections/appendices.

---

> > > ### Author Response · Authors · 2021-11-17
> > > **Response to Reviewer bLmw (3/5)**
> > >
> > > Here are the answers to further questions/comments:
> > >
> > > #Q4: overfitting doesn't seem to be a major problem in forecasting where model sizes are comparatively small (when compared with NLP for example), so this motivation doesn't convince me (page 1). A more convincing angle may be greater generality or speed.
> > >
> > > Indeed, the model sizes in time series forecasting are comparatively smaller to those used in NLP - at the same time, the datasets are also much smaller than those used in NLP, thus we cannot use NLP model sizes as a direct comparison for thinking about model sizes in time series forecasting. That being said, as mentioned above, large transformer-based models are unable to learn the seasonality/trend patterns due to the noise present in time series data, and proper inductive biases are required (i.e. Informer, as seen in Table 1 of main paper).
> > >
> > > #Q5:  page 2: "Firstly, we highlight that existing work using end-to-end deep forecasting methods directly model the time-lagged relationship along the observed data X" There is a considerable body of literature on probabilistic and deep forecasting models (e.g., https://iclr.cc/virtual/2021/spotlight/3409, but also Deep State Space Models at NeurIPS 20219 and much follow up work) which are in direct contradiction to this statement. In particular for multi-variate probabilistic and deep time series models, there is a growing body of literature which addresses the fact that Figure 2 is an over-simplification in the multi-variate context -- multiple X can have direct causal interaction (e.g., products that cannabilize/substitute/cross-sell)
> > >
> > > Thank you for highlighting this issue, we understand and are aware of the considerable body of literature in deep forecasting which model the multivariate interactions. We have corrected our writing to more accurately reflect this: “Firstly, we highlight that existing work using end-to-end deep forecasting methods, apart from modeling multivariate interactions, directly model the time-lagged relationship along the observed data $X$.”
> > >
> > > To clarify, we are trying to highlight the autoregressive nature of deep forecasting models combined with end-to-end training leading to capturing spurious correlations. Furthermore, figure 2 captures the multivariate interactions as each variable (including $X$) can be a vector, representing a multivariate time series and thus capture the multivariate interactions.
> > >
> > > #Q6: Are the embeddings trained on a single data set or across data sets?
> > >
> > > The embeddings are trained on a single dataset - the train split of the listed dataset (i.e. if we report results for ETTh1, then the embeddings are trained on the train split of ETTh1).
> > >
> > > #Q7: Over which time horizons do we train the embeddings and how do we forecast (rolling window evaluations), etc?
> > >
> > > Since the embeddings are trained in a self-supervised manner, the training process is independent of the forecasting horizon, i.e. the future timesteps (targets) are not needed in the pre-training phase, and thus the forecasting horizon is not relevant for this step.
> > >
> > > The downstream forecasting task uses the long time series forecasting evaluation setting as proposed in [1] - forecasts for the test set are made on a rolling window basis, but the prediction is made directly for the entire forecasting horizon (e.g. input lookback window to model -> directly predict 720 time steps).

---

> > > > ### Author Response · Authors · 2021-11-17
> > > > **Response to Reviewer bLmw (4/5)**
> > > >
> > > > #Q8: Why is MAE and MSE chosen when there are things like sMAPE and MASE which are more generally accepted?
> > > >
> > > > Our experiment setting follows a line of research [1, 2] which uses a normalized MSE and MAE for evaluation. Here, we include the sMAPE and ND, two popularly used forecasting metrics:
> > > >
> > > > Multivariate setting (Univariate is in the response below due to character limitation):
> > > >
> > > > | Methods |     | CoST  |       | TS2Vec |       | TNC    |       | MoCo   |       | Informer |       | TCN    |       |
> > > > |---------|-----|-------|-------|--------|-------|--------|-------|--------|-------|----------|-------|--------|-------|
> > > > | Metrics |     | sMAPE | ND    | sMAPE  | ND    | sMAPE  | ND    | sMAPE  | ND    | sMAPE    | ND    | sMAPE  | ND    |
> > > > | ETTh1   | 24  | 40.24 | 0.320 | 51.51  | 0.395 | 57.50  | 0.431 | 52.81  | 0.409 | 51.84    | 0.472 | 52.12  | 0.395 |
> > > > |         | 48  | 43.38 | 0.359 | 53.95  | 0.422 | 60.19  | 0.464 | 55.83  | 0.443 | 61.59    | 0.615 | 57.49  | 0.453 |
> > > > |         | 168 | 53.39 | 0.439 | 62.34  | 0.482 | 67.94  | 0.529 | 63.72  | 0.509 | 75.06    | 0.780 | 68.29  | 0.549 |
> > > > |         | 336 | 64.00 | 0.511 | 72.10  | 0.521 | 75.37  | 0.558 | 72.55  | 0.547 | 87.89    | 0.809 | 79.57  | 0.600 |
> > > > |         | 720 | 76.06 | 0.574 | 79.44  | 0.568 | 81.17  | 0.585 | 81.71  | 0.585 | 90.76    | 0.833 | 80.34  | 0.579 |
> > > > | ETTh2   | 24  | 73.78 | 0.245 | 71.29  | 0.239 | 74.75  | 0.270 | 71.32  | 0.240 | 75.83    | 0.293 | 85.64  | 0.389 |
> > > > |         | 48  | 80.40 | 0.314 | 77.28  | 0.301 | 81.58  | 0.329 | 76.67  | 0.291 | 104.15   | 0.638 | 91.21  | 0.458 |
> > > > |         | 168 | 95.31 | 0.418 | 94.38  | 0.472 | 100.63 | 0.521 | 94.47  | 0.450 | 109.89   | 0.779 | 107.18 | 0.705 |
> > > > |         | 336 | 96.08 | 0.443 | 98.65  | 0.529 | 105.27 | 0.590 | 98.94  | 0.525 | 105.03   | 0.661 | 105.35 | 0.648 |
> > > > |         | 720 | 94.69 | 0.461 | 103.67 | 0.633 | 104.44 | 0.671 | 101.56 | 0.597 | 110.83   | 0.981 | 105.90 | 0.594 |
> > > > | ETTm1   | 24  | 32.63 | 0.242 | 44.37  | 0.336 | 45.93  | 0.355 | 43.11  | 0.342 | 35.69    | 0.296 | 40.44  | 0.301 |
> > > > |         | 48  | 36.87 | 0.282 | 51.18  | 0.390 | 54.92  | 0.415 | 51.25  | 0.396 | 44.86    | 0.368 | 50.72  | 0.400 |
> > > > |         | 96  | 39.13 | 0.310 | 54.04  | 0.410 | 57.63  | 0.431 | 53.22  | 0.410 | 50.03    | 0.450 | 56.75  | 0.438 |
> > > > |         | 288 | 45.19 | 0.376 | 57.88  | 0.454 | 62.91  | 0.480 | 58.34  | 0.468 | 70.13    | 0.700 | 75.26  | 0.566 |
> > > > |         | 672 | 52.64 | 0.435 | 63.05  | 0.495 | 68.78  | 0.521 | 64.32  | 0.516 | 76.01    | 0.739 | 73.34  | 0.591 |
> > > > | Avg.    |     | 61.59 | 0.382 | 69.01  | 0.443 | 73.27  | 0.477 | 69.32  | 0.449 | 76.64    | 0.628 | 75.31  | 0.511 |
> > > >
> > > > As we can see, the rankings on sMAPE and ND are consistent with the original MSE/MAE metrics of this evaluation setting.
> > > >
> > > > [1] Zhou, Haoyi, et al. "Informer: Beyond efficient transformer for long sequence time-series forecasting." Proceedings of AAAI. 2021.
> > > >
> > > > [2] Yue, Zhihan, et al. "TS2Vec: Towards Universal Representation of Time Series." arXiv preprint arXiv:2106.10466 (2021).

---

> > > > > ### Author Response · Authors · 2021-11-17
> > > > > **Response to Reviewer bLmw (5/5)**
> > > > >
> > > > > Univariate setting:
> > > > >
> > > > > | Methods |     | CoST  |       | TS2Vec |       | TNC   |       | MoCo  |       | Informer |       | TCN   |       |
> > > > > |---------|-----|-------|-------|--------|-------|-------|-------|-------|-------|----------|-------|-------|-------|
> > > > > | Metrics |     | sMAPE | ND    | sMAPE  | ND    | sMAPE | ND    | sMAPE | ND    | sMAPE    | ND    | sMAPE | ND    |
> > > > > | ETTh1   | 24  | 41.06 | 0.280 | 40.91  | 0.276 | 43.10 | 0.338 | 39.39 | 0.277 | 52.42    | 0.476 | 49.55 | 0.444 |
> > > > > |         | 48  | 47.53 | 0.343 | 46.98  | 0.347 | 49.72 | 0.441 | 45.72 | 0.350 | 60.17    | 0.581 | 61.07 | 0.632 |
> > > > > |         | 168 | 54.01 | 0.443 | 59.62  | 0.560 | 58.93 | 0.617 | 54.75 | 0.496 | 55.03    | 0.513 | 78.95 | 1.096 |
> > > > > |         | 336 | 58.63 | 0.497 | 64.37  | 0.632 | 63.08 | 0.688 | 59.59 | 0.578 | 60.23    | 0.619 | 79.00 | 1.076 |
> > > > > |         | 720 | 60.10 | 0.617 | 66.98  | 0.710 | 68.96 | 0.824 | 64.11 | 0.702 | 54.44    | 0.476 | 82.31 | 1.190 |
> > > > > | ETTh2   | 24  | 21.64 | 0.167 | 23.60  | 0.184 | 24.75 | 0.192 | 24.29 | 0.188 | 23.55    | 0.180 | 26.85 | 0.213 |
> > > > > |         | 48  | 26.18 | 0.209 | 27.36  | 0.219 | 28.38 | 0.226 | 28.08 | 0.225 | 30.93    | 0.254 | 29.87 | 0.240 |
> > > > > |         | 168 | 33.81 | 0.277 | 34.38  | 0.289 | 34.62 | 0.289 | 34.96 | 0.294 | 39.68    | 0.351 | 36.24 | 0.309 |
> > > > > |         | 336 | 36.44 | 0.299 | 36.09  | 0.302 | 36.23 | 0.304 | 36.17 | 0.303 | 41.79    | 0.376 | 38.83 | 0.338 |
> > > > > |         | 720 | 37.05 | 0.316 | 36.48  | 0.314 | 36.46 | 0.315 | 36.46 | 0.314 | 40.64    | 0.371 | 36.77 | 0.318 |
> > > > > | ETTm1   | 24  | 27.50 | 0.162 | 28.42  | 0.170 | 30.28 | 0.189 | 28.09 | 0.169 | 32.71    | 0.195 | 33.32 | 0.207 |
> > > > > |         | 48  | 33.49 | 0.216 | 35.06  | 0.229 | 36.81 | 0.261 | 34.01 | 0.226 | 40.26    | 0.288 | 40.73 | 0.305 |
> > > > > |         | 96  | 39.29 | 0.271 | 41.76  | 0.294 | 42.32 | 0.328 | 39.28 | 0.278 | 57.29    | 0.555 | 48.00 | 0.414 |
> > > > > |         | 288 | 48.65 | 0.387 | 52.14  | 0.433 | 50.66 | 0.451 | 49.10 | 0.402 | 72.70    | 0.884 | 69.99 | 0.866 |
> > > > > |         | 672 | 54.10 | 0.482 | 58.66  | 0.552 | 55.91 | 0.543 | 55.42 | 0.505 | 74.97    | 0.935 | 77.22 | 1.047 |
> > > > > | Avg.    |     | 41.30 | 0.331 | 43.52  | 0.367 | 44.02 | 0.400 | 41.96 | 0.354 | 49.12    | 0.470 | 52.58 | 0.580 |

---

### Official Review · Reviewer_KrXv · 2021-11-02

**Correctness:** 4
**Technical Novelty And Significance:** 3
**Empirical Novelty And Significance:** 3
**Recommendation:** 6
**Confidence:** 4

**Details Of Ethics Concerns:**

No ethics concerns.

**Main Review:**

Strengths:
The idea of this paper seems novel and reasonable. Most parts are easy to follow. The experiments clearly show the improvement against baselines and the effectiveness of each model component.

Weaknesses:
1. The writing of the paper can be improved, e.g., several places can be more concise.
2. In the formulas of time domain and frequency domain contrastive losses, it is unclear why only one entry (time step) is selected as the anchor since both the trend and seasonality are concepts containing a series of time steps.
3. It is still unclear why representation learning methods can outperform end-to-end methods in such a large gap. It would be good if the authors can provide some insights and explanations about this.
4. When training the regression model on top of the learned backbone, are the parameters of the backbone frozen or not?
5. What is the specific setting of the TCN method, such as the model architecture, hyper-parameters, etc.


**Summary Of The Paper:**

The current training paradigm applied in most time series forecasting approaches jointly learns feature representations and the prediction function. This paper argues that this paradigm may lead to several issues, e.g., overfitting problems, obtaining false relations of the unpredictable noise in the data, and entangling representations.

To tackle these challenges, this paper proposes a new framework (CoST) to first learn error-free feature representations, and then fine-tune the representations through a simple regressor. In the feature learning stage, a pretext task is constructed by using data augmentation and contrastive learning. The goal is to learn disentangled seasonal and trend representations. Moreover, time domain and frequency domain contrastive losses are incorporated to learn discriminative trends and seasonal representations, respectively.


**Summary Of The Review:**

This paper still has space to improve. I will preserve my score before reading the authors' responses.

---

> ### Author Response · Authors · 2021-11-17
> **Response to Reviewer KrXv**
>
> Thank you for your valuable comments and feedback. Below are our responses to the issues raised:
>
> #Q1:  In the formulas of time domain and frequency domain contrastive losses, it is unclear why only one entry (time step) is selected as the anchor since both the trend and seasonality are concepts containing a series of time steps.
>
> For the frequency domain contrastive loss, all entries (in this case, frequencies, since a discrete Fourier transform is applied), are used for the contrastive loss. This is indicated by the loss function summing over index i=0,..,F.
>
> For the time domain contrastive loss, a single time step of representation, firstly, is the output of the mixture of autoregressive experts - meaning that it is already a representation of a series of time steps. Next, the backbone encoder also has a large receptive field which encodes information from multiple timesteps. Finally, we opt to use only a single entry due to implementation reasons  - it is simpler to control the number of anchors if it is dependent only on the batch size, rather than being dependent on both the batch size, and the number of time steps. Overall, there would not be much of a difference for both formulations, apart from implementation details.
>
> #Q2: It is still unclear why representation learning methods can outperform end-to-end methods in such a large gap. It would be good if the authors can provide some insights and explanations about this.
>
> The reasons to why the two-stage approach is able to beat existing end-to-end approaches comes in two parts:
>
> 1. Contrastive learning is able to learn robust representations via interventions on the error variable, whereas end-to-end approaches tend to overfit on spurious correlations. In the domain of CV, contrastive learning has matched the performance of supervised learning methods, but still has not beaten them. However, when dealing with time series, the data is noisier and more non-stationary. Thus, the robust representations from contrastive learning brings about a larger improvement in performance than in CV.
>
> 2. Our seasonal-trend representations are a powerful inductive bias for time series forecasting. Our STL-based approach is different from existing forecasting approaches - we emphasize that our seasonal-trend representations lies in the latent space, while traditional STL based methods do it in the input space. CoST has a much stronger expressive capability to uncover complex seasonal and trend patterns. Furthermore, we argue that a large transformer-based model with sufficient parameters is unable to learn such patterns because time series data is very noisy and it is easy for a large model to learn spurious autocorrelations - incorporating the proper inductive bias is crucial to help the model generalize.
> We will clarify this in the final version.
>
> #Q3: When training the regression model on top of the learned backbone, are the parameters of the backbone frozen or not?
>
> Yes, the parameters of the backbone are frozen.
>
> #Q4: What is the specific setting of the TCN method, such as the model architecture, hyper-parameters, etc.
>
> For the TCN, we follow the model architecture described in [1] which is included as a baseline in our results. It is described in their paper:
>
> “In the encoder of TS2Vec, the linear projection layer is a fully connected layer that maps the input channels to hidden channels, where input channel size is the feature dimension, and the hidden channel size is set to 64. The dilated CNN module contains 10 hidden blocks of ”GELU → DilatedConv → GELU → DilatedConv” with skip connections between adjacent blocks. For the i-th block, the dilation of the convolution is set to 2 i . The kernel size is set to 3. Each hidden dilated convolution has a channel size of 64. Finally, an output residual block maps the hidden channels to the output channels, where the output channel size is the representation dimension.” The representation dimensions are set to 320. We have included these details in Appendix E of the updated submission.
>
> [1] Yue, Zhihan, et al. "TS2Vec: Towards Universal Representation of Time Series." arXiv preprint arXiv:2106.10466 (2021).

---

> > ### Comment · Reviewer_KrXv · 2021-11-25
> > **Response to the authors**
> >
> > Dear Author(s),
> >
> > Thank you much for addressing my concerns. After reading the authors' responses and other reviewers' comments, I agree with the technical contributions of this paper and would like to upgrade my score to 6.
> >
> > By the way, I would like to check two more details:
> > 1. Why do the authors employ the synthetic dataset for the visualization instead of the real-world datasets used in the experiments?
> > 2. In Sec 1, "the situation is exacerbated ..... multiple local independent modules of the data-generating process – and a local independent module experiences a distribution shift." After reading the whole paper, I am still not sure what is the local independent module here. Could you please explain more details here?
> >
> >
> > --Reviewer KrXv

---

> > > ### Author Response · Authors · 2021-11-26
> > > **Response to additional clarifications**
> > >
> > > Thank you for the update, and here are our responses to your further clarifications,
> > >
> > > 1. The goal of the case study visualization is to visualize the clusterability/separability of different trend and seasonality patterns. Thus, we need to construct a synthetic dataset in which we know the data-generating process, as well as the respective trend and seasonality patterns of each time series and are able to label them for visualization. We do not have such labels and information for the real-world datasets.
> > >
> > > 2. The local independent module refers to the trend and seasonality being independent causal mechanisms [1,2]. The independent causal mechanism assumption states that the causal generative process is composed of autonomous modules that do not inform or influence each other. By applying this assumption to time series data, we assume that shifts in trend do not inform or influence the seasonality patterns, and vice versa. This motivates the learning of disentangled seasonal-trend representations as well as representations robust to interventions on the error component. We elaborate about this in the main paper on page 3, with the paragraph starting "Secondly, by the independent mechanisms assumption ...".
> > >
> > > [1] Schölkopf, Bernhard, et al. "Toward causal representation learning." Proceedings of the IEEE 109.5 (2021): 612-634.
> > >
> > > [2] Parascandolo, Giambattista, et al. "Learning independent causal mechanisms." International Conference on Machine Learning. PMLR, 2018.

---

### Official Review · Reviewer_S8vv · 2021-11-02

**Correctness:** 4
**Technical Novelty And Significance:** 3
**Empirical Novelty And Significance:** 3
**Recommendation:** 6
**Confidence:** 4

**Main Review:**

## Strengths
1. The disentanglement of trends and seasonalities is well-motivated. The interpretation of invariance under intervention on E is valid.
2. The introduced contrastive losses is a good proxy to learn discriminative representations.
3. The extensive empirical results show the effectiveness of the proposed method and the ability of identifying multiple patterns.

## Weakness
1. The method of making forecasts from learned representations is not presented, making it incomplete for reproducing the results. In addition, a joint loss with the forecasting error should also be compared with the current contrastive training.
2.  In case study, besides clustering of multiple patterns, the disentanglement of trends and seasonalities should also be verified. For example, the representation of the same seasonal pattern is supposed to be overlapped even mixed with different trended patterns.

## Comments and Concerns
1. In ablation study, what is the difference between TFD and MARE? The "Trend Feature Disentangler" paragraph in page.4 states TFD is a mixture of AR experts.
2. In the selected datasets, the seasonalities of samples are highly similar (e.g. daily and weekly) The negative samples in frequency contrastive loss may be indistinguishable from the positive one in the first place. It may hurt the effectiveness of the frequency representations.
3. Minor: $d$ should be $m$ in the sec 2. "Problem Formulation" $\hat{X}\in\mathbb{R}^{k\times d}$?

**Summary Of The Paper:**

This paper introduce a framework of learning disentangled trend and seasonal representations of time series and its application to forecasting tasks. It employs contrastive losses to distill discriminative trend and seasonal representations. DFT is used in obtaining frequency domain information. Empirical results show the proposed framework is able to outperform end-to-end trained forecasters and other representation learning methods. In addition, the learned representations can be clustered to distinguish multiple trends or seasonalities.

**Summary Of The Review:**

Overall I think this paper is well-written with clear motivation and technical path. It is acceptable but can be further improved in several aspects.

---

> ### Author Response · Authors · 2021-11-17
> **Response to Reviewer S8vv (1/2)**
>
> #Q1: Making forecasts from learned representations is not presented, a joint loss with the forecasting error should be compared.
>
> Thank you for the valuable feedback! To address your concerns regarding the method of obtaining forecasts from learned representations, self-supervised learning approaches are first trained on the train split, and a ridge regression model is trained on top of the learned representations to directly forecast the entire prediction length. We include this in paragraph “Evaluation Setup” of Section 4.1.
>
> Here are the results of a joint loss with the forecasting error (denoted CoST$^\dagger$ (w/ Contrastive Loss)). The methods denoted with $\dagger$ are approaches trained end-to-end with a forecasting loss. We hypothesise that the reason for the lower performance of the joint loss with forecasting error could be due to the forecaster being trained with the additional noise being used for the contrastive learning phase.
>
> |                                         | TFD          | MARE         | SFD          | LFL          | FDCL         | Multivariate |       | Univariate |       |
> |-----------------------------------------|--------------|--------------|--------------|--------------|--------------|--------------|-------|------------|-------|
> |                                         |              |              |              |              |              | MSE          | MAE   | MSE        | MAE   |
> | Trend                                   | $\checkmark$ |              |              |              |              | 0.882        | 0.674 | 0.115      | 0.243 |
> |                                         | $\checkmark$ | $\checkmark$ |              |              |              | 0.789        | 0.630 | 0.105      | 0.235 |
> | Seasonal                                |              |              | $\checkmark$ |              | $\checkmark$ | 0.905        | 0.675 | 0.105      | 0.237 |
> |                                         |              |              | $\checkmark$ | $\checkmark$ |              | 0.895        | 0.721 | 0.103      | 0.239 |
> |                                         |              |              | $\checkmark$ | $\checkmark$ | $\checkmark$ | 0.862        | 0.668 | 0.129      | 0.255 |
> | CoST$^{\dagger}$ (w/o Contrastive Loss) |              |              |              |              |              | 1.376        | 0.834 | 0.228      | 0.366 |
> | CoST$^{\dagger}$  (w Contrastive Loss)  |              |              |              |              |              | 1.477        | 0.909 | 0.965      | 0.883 |
> | MoCo                                    |              |              |              |              |              | 0.996        | 0.721 | 0.112      | 0.248 |
> | SimCLR                                  |              |              |              |              |              | 1.021        | 0.730 | 0.113      | 0.248 |
> | CoST                                    | $\checkmark$ | $\checkmark$ | $\checkmark$ | $\checkmark$ | $\checkmark$ | 0.781        | 0.625 | 0.102      | 0.233 |
>
> #Q2:  Case study of the disentanglement of trends and seasonalities
>
> Thanks for your suggestion. We have updated the submission with an additional Appendix I, Case Study: Disentanglement to showcase this disentanglement. Please view this section for the additional figures.
>
> To exhibit the disentanglement in the case study, in Figure 5a, by plotting the T-SNE representations of the trend component and the seasonality components separately, the trend representations indeed have better clusterability and the seasonality representations have a degree of overlap between the two trend patterns
>
> For the seasonality components, we plot the T-SNE visualizations, in Figure 5b, of the trend and seasonal components of CoST separately, and as seen above, the trend representations have a higher degree of overlap/mixing between the three seasonality patterns than the seasonality representations. Here, we use complex seasonalities to highlight the stronger capability of seasonality representations in extracting seasonal patterns.

---

> > ### Author Response · Authors · 2021-11-17
> > **Response to Reviewer S8vv (2/2)**
> >
> > Answers to other clarifications are below:
> >
> > #Q3: In ablation study, what is the difference between TFD and MARE? The "Trend Feature Disentangler" paragraph in page 4 states that TFD is a mixture of AR experts.
> >
> > TFD without MARE refers to the module which only uses a single AR expert, with the largest kernel size, h//2. TFD with MARE refers to the module which uses the mixture of AR experts as proposed in Section 3.1.
> >
> > #Q4: In the selected datasets, the seasonalities of samples are highly similar (e.g. daily and weekly) The negative samples in frequency contrastive loss may be indistinguishable from the positive one in the first place. It may hurt the effectiveness of the frequency representations.
> >
> > Thank you for the insights. Seeking better sampling strategies is a general problem in SSL, and has been actively investigated in CV recently. Albeit for a different case, we do find that different datasets and experiment settings have a different optimal hyperparameter in the comparison between the trend and seasonality contrastive losses (Table 2 in the main paper). However, selecting the correct seasonality to focus on (without prior knowledge) as a hyperparameter would be too costly, since there are too many frequencies to perform a hyperparameter search for the appropriate weighting. We agree how to adaptively and automatically select the proper contrastive loss is an interesting and important problem, and leave this investigation for future work.
> >
> > #Q5: Minor: d should be m in the sec 2. "Problem Formulation"
> >
> > Thank you for spotting this issue, we have made the appropriate changes.
> >
> > Finally, we will be adding the additional experiments and updates to the case study in the appendix in the final submission, as well as clarifying the differences between TFD and MARE.

---

> > > ### Author Response · Authors · 2021-11-29
> > > **Any other concerns need us to address?**
> > >
> > > We thank the reviewer for the valuable comments. As the discussion phase will end soon, we would like to check if our response has addressed your concern. If the reviewer has any further questions, we would be happy to take a look. Thank you.

---

> > > > ### Comment · Reviewer_S8vv · 2021-11-29
> > > > **Thank you for your response**
> > > >
> > > > The response addressed all my concern and I would be happy to see it get accepted. But due to the limitation as Q4 states, I will keep my score as it is and hope to see it improved in the future.

---

### Official Review · Reviewer_97uN · 2021-11-06

**Correctness:** 4
**Technical Novelty And Significance:** 3
**Empirical Novelty And Significance:** 3
**Recommendation:** 8
**Confidence:** 3

**Main Review:**

Strength:
- The idea of  disentangling trend and seasonal features and using the learnt representations for forecasting is novel.
- CoST was compared to different representation learning and end-to-end learning appraoches on 5 datasets and showed strong empirical results.
- The paper preformed ablation studies induding investigating the use of different backbone encoders, the use of different regressors for forecasting and the effect of different componets of CoST on accuracy.
- Through synthetic datasets the paper showsthat  CoST can actually disentangle seasonal and trend features.

Weakness:
- Missing simple baselines, like for example linear regression on the raw features rather than the learned representations and linear regression on the features outputted from the backbone encoder.
- Some issues with paper writting please see clearification questions below.
- Missing analysis of computational resourses and training time needed by CoST when compared to other methods.

Clarification questions:

- In figure3 shouldn't V(l) and V(g) be V(T) and V(S).
- Is it possible to provide the code to verify the approach and experiments?
- For the multivariate time series are you forecasting all features or just using them as input and forecasting target features.
- What does the second half of table 2 show (the detailed breakdown for multivaritate setting).
- For tables 3-5 is this the mean accross all ETT datasets?

**Summary Of The Paper:**

The paper proposes a contrastive learning framework for time series forecasting CoST. This is done by detangling trend and seasonal features than using the learnt feature representations with a simple regression. Trend feature disentangler consists of a mixture of autoregressive experts followed by average pooling. While the seasonal feature is done by changing the features into the frequency domain via FFT and then applying a linear layer with weights to every different frequency. The overall loss is the weighted time-domain contrastive loss from trend feature disentangler and frequency domain contrastive loss which contains both phase and amplitude loss from seasonal feature disentangler. The proposed method was evaluated on 5 datasets in both a multivariate and univariate setting. CoST was compared against different representation learning and end-to-end learning appraoches.

**Summary Of The Review:**

The overall idea is novel and shows strong empirical results.
The paper performs ablation studies justifying different design choices.
The paper shows the ability of the proposed method to disentangle trends and seasonal features through synthetic datasets.

---

> ### Author Response · Authors · 2021-11-17
> **Response to Reviewer 97uN (1/2)**
>
> Thank you for the positive review and insightful comments. We have performed the additional requested baseline and will include these results in the final version.
>
> #Q1: Missing the baseline of linear regression on raw time series
>
>  Overall, we can see that CoST is significantly better than  linear regression on both univariate and multivariate settings.
>
> Additional baseline of linear model on raw time series (averaged over the 3 ETT datasets):
>
> | Methods      |  | CoST  |       | Informer |       | LogTrans |       | TCN   |       | LSTnet |       | Linear |       |
> |--------------|--|-------|-------|----------|-------|----------|-------|-------|-------|--------|-------|--------|-------|
> | Metrics      |  | MSE   | MAE   | MSE      | MAE   | MSE      | MAE   | MSE   | MAE   | MSE    | MAE   | MSE    | MAE   |
> | Multivariate |  | 0.781 | 0.625 | 1.342    | 0.859 | 1.635    | 1.053 | 1.385 | 0.834 | 2.411  | 1.782 | 2.337  | 0.820 |
> | Univariate   |  | 0.102 | 0.233 | 0.210    | 0.362 | 0.228    | 0.391 | 0.230 | 0.372 | 1.242  | 0.536 | 1.370  | 0.379 |
>
> #Q2: Computation resources and training time analysis.
>
> | Phase     | H   | CoST          | TS2Vec       | TNC             | MoCo         | Informer | TCN    |
> |-----------|-----|---------------|--------------|-----------------|--------------|----------|--------|
> | Training  | 24  | 262.78 + 7.19 | 91.9 + 5.45  | 1801.58 + 4.78  | 31.3 + 6.57  | 331.32   | 108.36 |
> |           | 48  | 262.78 + 8.41 | 91.9 + 6.44  | 1801.58 + 5.79  | 31.3 + 8.03  | 167.18   | 109.30 |
> |           | 96  | 262.78 + 10.0 | 91.9 + 8.13  | 1801.58 + 7.23  | 31.3 + 9.40  | 325.51   | 110.02 |
> |           | 288 | 262.78 + 19.9 | 91.9 + 16.47 | 1801.58 + 14.51 | 31.3 + 17.83 | 449.86   | 112.08 |
> |           | 672 | 262.78 + 38.3 | 91.9 + 32.22 | 1801.58 + 28.21 | 31.3 + 36.63 | 587.25   | 112.87 |
> | Inference | 24  | 23.60 + 0.04  | 3.73 + 0.05  | 3.38 + 0.05     | 3.67 + 0.07  | 10.32    | 3.20   |
> |           | 48  | 23.60 + 0.07  | 3.73 + 0.06  | 3.38 + 0.09     | 3.67 + 0.08  | 5.78     | 3.67   |
> |           | 96  | 23.60 + 0.11  | 3.73 + 0.08  | 3.38 + 0.09     | 3.67 + 0.10  | 11.32    | 4.78   |
> |           | 288 | 23.60 + 0.24  | 3.73 + 0.18  | 3.38 + 0.21     | 3.67 + 0.22  | 17.19    | 7.19   |
> |           | 672 | 23.60 + 0.34  | 3.73 + 0.30  | 3.38 + 0.33     | 3.67 + 0.33  | 25.62    | 11.93  |
>
> The table above shows the runtime in seconds for each phase, for representation methods, A + B, where A refers to the time for the encoder, and B is the time for the ridge regressor in the downstream phase. All experiments are performed on an NVIDIA A100 GPU. Do note that for Informer, we follow the hyperparameters as described by the authors which may vary for different forecasting horizons, thus the runtime may not be increasing as the forecasting horizon increases. We want to highlight that for all representation learning approaches, the ridge regressor portions should be equal since the dimension size used are the same across all methods, any differences are simply due to randomness. Despite a slightly higher training time compared to some of the baseline approaches, CoST achieves much better results (refer to Table 1 in the main paper). Furthermore, the extra computation time of CoST as compared to TS2Vec is due to the sequential computation of each expert in the mixture of AR expert component, it can be further accelerated by parallel methods [1].
>
> [1] He, Jiaao, et al. "FastMoE: A Fast Mixture-of-Expert Training System." arXiv preprint arXiv:2103.13262 (2021).
>
>
> Asymptotic Complexity: CoST uses the same backbone encoder as TS2Vec, with additional parameters for the Trend Feature Disentangler and Seasonal Feature Disentangler modules. The Trend Feature Disentangler requires O(N * K * D^2) more parameters, where N is the number of AR experts, K is the largest order of AR, and D is the feature dimensionality. The Seasonal Feature Disentangler requires O(T * D^2) more parameters where T is the lookback window length.

---

> > ### Author Response · Authors · 2021-11-17
> > **Response to Reviewer 97uN (2/2)**
> >
> > Answers to other clarifications are below:
> >
> > #Q3: In figure3 shouldn't V(l) and V(g) be V(T) and V(S).
> >
> > Thank you for spotting this issue, we have made the appropriate changes.
> >
> > #Q4: Is it possible to provide the code to verify the approach and experiments?
> >
> > We have uploaded our anonymized code as supplementary material.
> >
> > #Q5: For the multivariate time series are you forecasting all features or just using them as input and forecasting target features.
> >
> > In the multivariate setting, all features are used as both input and forecasting targets.
> >
> > #Q6: What does the second half of table 2 show (the detailed breakdown for multivariate setting).
> >
> > The detailed breakdown is meant to highlight that while a lower value of alpha is preferred on average across the datasets used for the sensitivity analysis, for certain datasets/experiment settings, a larger value of alpha is preferred - leading to our choice of selecting an intermediate value of alpha for the main experiment results.
> >
> > #Q7: For tables 3-5 is this the mean across all ETT datasets?
> >
> > Yes, the results presented in tables 3, 4, 5 are the mean across ETTh1, ETTh2, and ETTm1.

---

> > > ### Comment · Reviewer_97uN · 2021-11-19
> > > **Thank you for your respond**
> > >
> > > Thank you for your respond.
> > > Based on the  respond I will leave my score as is.
> > > Please add the M5 dataset experiments to your code base.

---

> > > > ### Author Response · Authors · 2021-11-22
> > > > **Updated Supplementary Materials with M5 dataset**
> > > >
> > > > We have updated the supplementary materials with the M5 dataset.

---

### Author Response · Authors · 2021-11-22
**General Response To All Reviewers**

We thank all reviewers for their valuable comments and insightful suggestions and will incorporate all feedbacks into the final version of our work. Accordingly, we have revised the paper and conducted extensive additional experiments as suggested by all Reviewers.  We summarize our major efforts during this phase as follows:

- We performed further experiments on the requested baselines (N-BEATS, DeepAR and Linear model on raw time series) and dataset (M5).
- We performed further ablation studies about: 1. the joint loss with the forecasting error, 2. using the embeddings as co-variates for an end-to-end learning of forecasting model.
- We provided the training time and computation resources analysis.
- We provided the experimental results in terms of sMAPE and ND, two popular scaled-based evaluation metrics.
- We performed the requested case study of the the disentanglement of trends and seasonalities.
- We clarified the benefits of our proposed two-stage approach over the existing end-to-end approaches.
- We modified some expressions in the original paper to improve clarity, and corrected typos in the paper.

In the following, we will address the concerns from each Reviewer individually.

---

> ### Comment · Reviewer_KrXv · 2021-11-25
> **Response to Authors**
>
> Thank you for your revision. The new version seems to improve a lot.

---

### Decision · Program_Chairs · 2022-01-20

**Decision:**

Accept (Poster)

**Comment:**

The paper proposes to learn disentangled trends and seasonal representations of time series for forecasting tasks. It shows separating the representation learning and downstream forecasting task to be a more promising paradigm than the standard end-to-end supervised training approach for time-series forecasting.

During the post-rebuttal phase, there were interactions from all the reviewers, and reviewer KrXv raised the score. The reviewers think the contrastive learning method is novel and the added experiments have strengthened the paper.  The authors are encouraged to include more standard datasets (M5) in the final version.

Based on the above reasons, I am recommending accepting this paper.